# Junction Mapper is a novel computer vision tool to decipher cell–cell contact phenotypes

Helena Brezovjakova[1†], Chris Tomlinson[2†], Noor Mohd Naim[1‡], Pamela Swiatlowska[1], Jennifer C Erasmus[1], Stephan Huveneers[3], Julia Gorelik[1], Susann Bruche[1§*], Vania MM Braga[1*]

[1]National Heart and Lung Institute, National Institutes of Health, London, United Kingdom; [2]Bioinformatics Data Science Group, Faculty of Medicine, Imperial College London, London, United Kingdom; [3]Department Medical Biochemistry, Amsterdam Cardiovascular Sciences, Amsterdam UMC, University of Amsterdam, Amsterdam, Netherlands

*For correspondence:
susann.bruche@dpag.ox.ac.uk
(SB);
v.braga@imperial.ac.uk (VMMB)

[†]These authors contributed equally to this work

Present address: [‡]PAPRSB Institute of Health Sciences, Universiti Brunei, Darussalam, Brunei; [§]Department of Anatomy, Physiology and Genetics, Oxford University, Oxford, United Kingdom

Competing interests: The authors declare that no competing interests exist.

**Abstract** Stable cell–cell contacts underpin tissue architecture and organization. Quantification of junctions of mammalian epithelia requires laborious manual measurements that are a major roadblock for mechanistic studies. We designed Junction Mapper as an open access, semi-automated software that defines the status of adhesiveness via the simultaneous measurement of pre-defined parameters at cell–cell contacts. It identifies contacting interfaces and corners with minimal user input and quantifies length, area and intensity of junction markers. Its ability to measure fragmented junctions is unique. Importantly, junctions that considerably deviate from the contiguous staining and straight contact phenotype seen in epithelia are also successfully quantified (i.e. cardiomyocytes or endothelia). Distinct phenotypes of junction disruption can be clearly differentiated among various oncogenes, depletion of actin regulators or stimulation with other agents. Junction Mapper is thus a powerful, unbiased and highly applicable software for profiling cell–cell adhesion phenotypes and facilitate studies on junction dynamics in health and disease.

## Introduction

Tight contacts with neighbours to form a cohesive sheet of cells is a fundamental property of multi-cellular organisms and underpins organ development and function. Conversely, signalling pathways necessary to maintain junctions are often targeted by pathogens and underlie key mechanisms of diseases of the vasculature, heart and different epithelial organs.

Attachment to neighbouring cells has a distinct configuration in different cell types and is dynamically remodelled in homeostasis and diseases. In epithelial tissues, the characteristic cell–cell adhesion site appears as a straight and tense or stiff junction, represented by an apparent contiguous, adjoining staining of E-cadherin receptors (the epithelia-specific cadherin protein). Following stimuli such as growth factor treatment or oncogene expression in epithelial cells, the dynamic nature of cell–cell contacts is manifested in a variety of ways: disturbances in the configuration of the contacting interface between cells, fragmentation of cadherin staining or thinning out of the distribution of receptors from contacting cell borders (*Braga et al., 2000*; *Erasmus et al., 2016*; *Frasa et al., 2010*; *Lozano et al., 2008*; *Nola et al., 2011*).

However, a linear junction appearance does not apply to other cell types that also require strong cell–cell attachment. Intercalated discs, specialised junctions in cardiomyocytes, sustain considerable mechanical stress with heart beating (*Ehler, 2016*; *Vermij et al., 2017*). Extensive remodelling of

the intercalated discs composition (*Estigoy et al., 2009*) and architecture is observed in cardiac 'aging (*Sessions and Engler, 2016*; *Tribulova et al., 2015*), diabetes-induced cardiopathies (*Adeghate and Singh, 2014*), arrythmogenic cardiomyopathies (*Calore et al., 2015*) and cardiac hypertrophy and failure (*Lyon et al., 2015*). During vascular homeostasis, endothelial cell–cell contacts may have a similar appearance as epithelial junctions (*Dejana and Orsenigo, 2013*; *Malinova and Huveneers, 2018*). Upon stimulation with inflammatory agonists (*Radeva and Waschke, 2018*), endothelial contacts undergo adjustments to increase permeability, changing from a linear to a zig-zag configuration and the appearance of gaps between cells (*Malinova and Huveneers, 2018*). Collectively, the above data demonstrate that distinct patterns of junctions are stimulus-dependent and reflect the specific destabilization (or strengthening) of cadherin receptors at contact sites in various cell types.

Despite the extensive scientific progress in our understanding of how cell–cell contacts are modulated, how these distinct phenotypes of junction modulation are fully attained is still unclear. A major road block to furthering mechanistic studies on junction regulation is the restricted capability and efficiency of the quantitative image analysis currently available. Existing imaging platforms (i.e. Cell Profiler) (*Carpenter et al., 2006*; *McQuin et al., 2018*) are fantastic resources for cell biologists. For *Drosophila* or nematode epithelia, several software are available for quantification of morphogenesis that robustly detect cell boundaries during morphogenesis (i.e. during dorsal closure or germ band extension) (*Curran et al., 2017*; *Sumi et al., 2018*). In contrast, for mammalian epithelial cells, available systems are not always suitable for the precise delineation of cell borders and the output of current pipelines is mostly morphometric parameters (i.e. cell size, shape, number or texture) (*Campbell et al., 2017*; *McQuin et al., 2018*; *Yin et al., 2013*). Cell–cell borders of mammalian cells are particularly difficult to detect because of cytoplasmic noise, irregular shapes (*Held et al., 2011*) and variable junction phenotypes, particularly when junctions are severely disrupted. This hitherto prevents an objective approach to analyse regulation of cell–cell contacts of mammalian cells and tissues.

Nevertheless, previous computer vision studies of mammalian junctions report the automated quantification of a single heterotypic junction (e.g. tumour-immune cell contact or host-pathogen contact) (*Graus et al., 2014*; *Merouane et al., 2015*), morphometry of mammary gland spheroids (*Härmä et al., 2014*) or dynamics of VE-cadherin contacts during cell rearrangements in angiogenesis (*Bentley et al., 2014*). Disruption of cell–cell contacts has been assessed in high-throughput manner by coupling junction segmentation with cell tracker and endothelia stimulation (*Seebach et al., 2015*), cadherin intensity at junctions (*Erasmus et al., 2016*) or indirectly, by increased inter-nuclear distance as cells scatter (*Loerke et al., 2012*).

Notwithstanding these successful studies, available methodology does not enable quantification of distinct patterns of organization of receptors or junction morphometry that are readily identified by the human eye. In addition, manual methods available for junction quantification rely on intensity levels and thresholding, which are not appropriate to detect junction attributes such as alterations in shape, length, fragmentation or continuity of cell–cell contacts. Non-intensity-based attributes of junctions are usually defined visually and/or painstakingly analysed via laborious user-dependent quantification of individual junctions. For example, the switch between a straight to undulated cell–cell contact occurs without apparent changes in receptors levels at contact sites (*Otani et al., 2006*). In this case, rather than alterations of receptor levels at junctions, it involves impaired signalling of the small GTPase Cdc42 to modulate the amount of contraction, making cell–cell contacts less stiff and tense (*Oda et al., 2014*; *Otani et al., 2006*).

To address the above issues, we developed a semi-automated pipeline, Junction Mapper, which can fully capture the distinct patterns of junction perturbation by diverse stimuli in various cell types. It can efficiently (i) identify cell boundaries and cell–cell corners, (ii) describe phenotypes of junction architecture and (iii) quantify parameters that reflect the distribution and organization of junctional markers along the contacting interface. To broaden the software suitability to different models, we validate the robustness of the Junction Mapper software in endothelial cells and cardiomyocytes that show distinct receptor organization and architecture when compared to epithelial cells. The repertoire of parameters distinguishes subtle differences of junction disruption and provides a fingerprint for each stimulus, with insights into modes of action and how efficient and functional junctions are.

We envisage that the analytical capabilities of Junction Mapper will be invaluable for the scientific community to perform quantitative image analysis in mechanistic and translational studies of cell–cell contacts. Most importantly, the generation of tools to facilitate unbiased phenotype identification will be a major step forward to understand how junction dynamics are modulated in homeostasis and pathologies of different tissues.

## Results

In normal epithelia, a junction between neighbouring cells usually appears as a straight, taut line, with E-cadherin receptors uniformly distributed along the contacting interface (*Figure 1A*). Junctions are delimited by corners between three or more cells, where a specialised type of contacts are formed (tricellular junctions) (*Figure 1A*). Distinct stimuli disrupt the above junction architecture in different ways, from minor reduction in levels to complete removal of adhesion receptors from contacting cells (*Figure 1B*). Concomitant with changes in levels, junction configuration and architecture are also compromised, which are not always captured by intensity measurements.

We designed and validated a semi-automated system (*Figure 2*), Junction Mapper, that builds from our quantitative analysis of images from RNAi screens (*Erasmus et al., 2016*). Our previous software defines an E-cadherin mask to calculate the intensity specifically around junctions as a

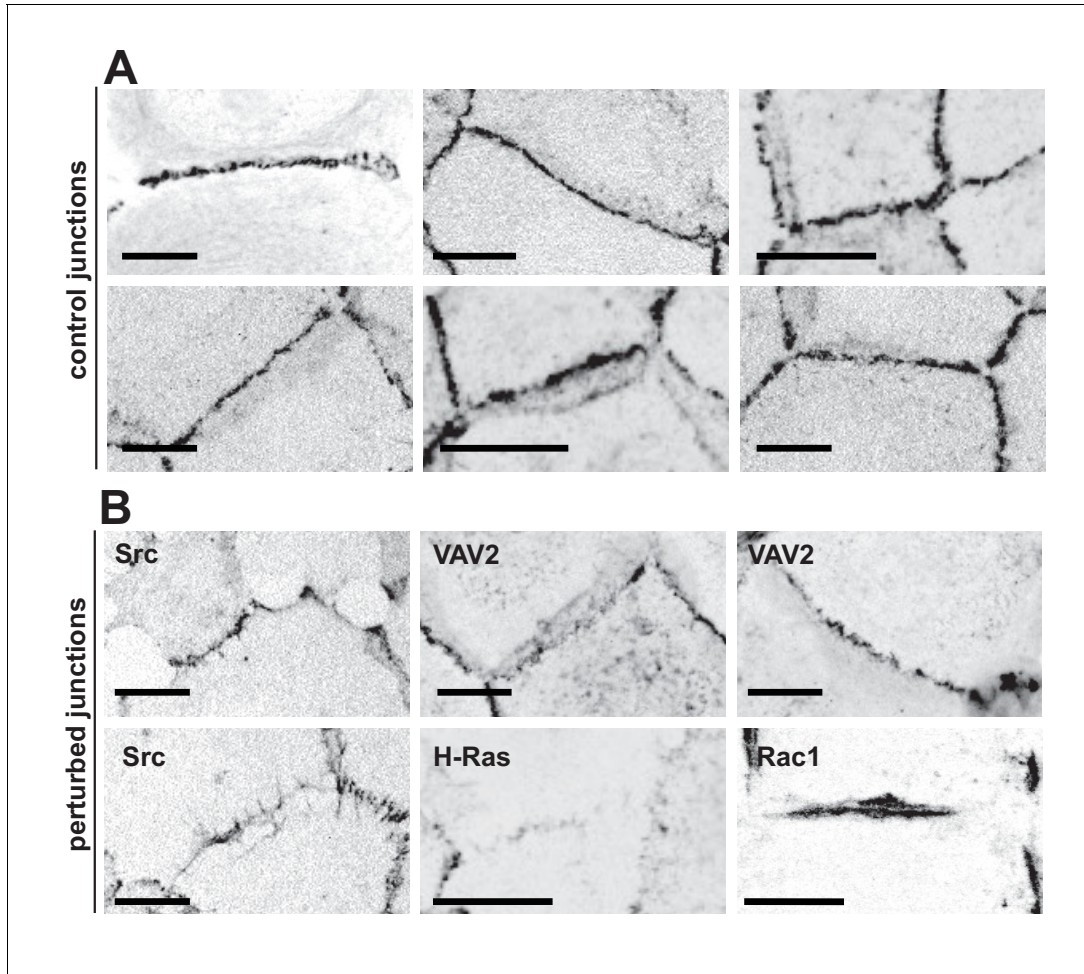

**Figure 1.** Appearance of normal and perturbed junctions. (**A**) Representative images of junctions from normal epithelial cells, which are characterized by cell–cell contacts between neighbouring cells appearing as a straight line, with E-cadherin receptors uniformly distributed along the contacting interface. Junctions are delimited by corners between three or more cells, where a specialised type of contact is formed (tricellular junctions). (**B**) Distinct stimuli disrupt the above junction architecture in different ways, from minor reduction in levels to complete removal of adhesion receptors from contacting cells. Concomitant with changes in receptor levels, junction configuration and architecture are also compromised, alterations which are not always captured by intensity measurements. Scale bars = 10 μM.

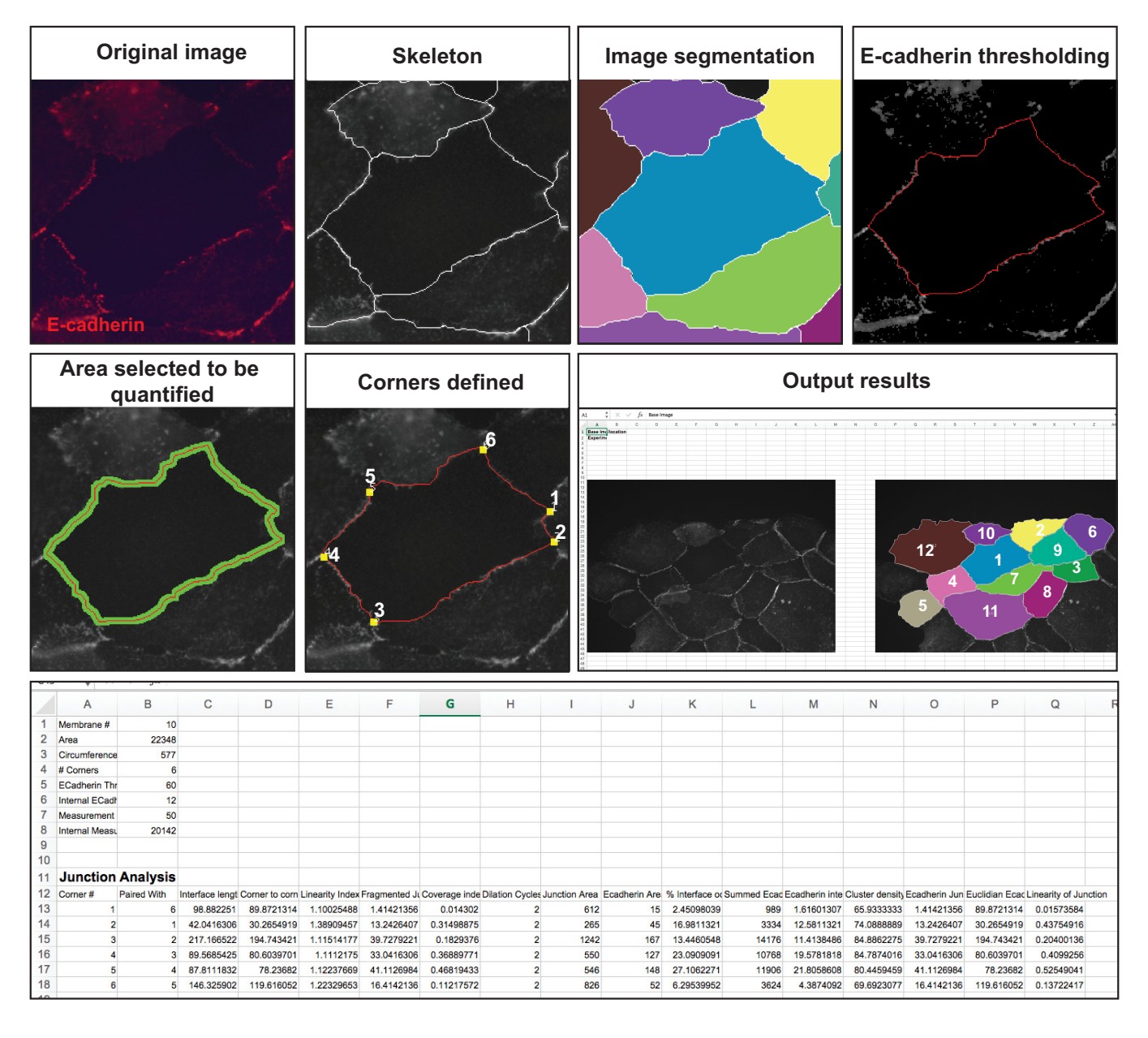

**Figure 2.** Junction Mapper – overview of quantification process. The original grey scale image stained for a junctional marker (E-cadherin) is uploaded in the program, which identifies the edge map of each cell semi-automatically to define the cell boundaries onto which measurements are made (skeleton, one pixel wide). This is superimposed onto the images to allow correction of missing boundaries and small errors interactively by the user. The image is then segmented to identify each cell to be quantified. A threshold is set to remove non-specific staining, and the skeleton is dilated (pixels) to select the area to be quantified that includes all staining at cell–cell contacts. Finally, cell corners are defined automatically or manually. Thresholded images are then quantified using the dilated mask and results are shown in Excel files as individual measurements of specific junctions as defined by the pairing of two different corners. Detailed steps are described in *Figure 2—figure supplement 1* and Appendix 1.

The online version of this article includes the following source data and figure supplement(s) for figure 2:

**Figure supplement 1.** Summary of the automatic detection of cell–cell borders by Junction Mapper and user-controlled adjustments.
**Figure supplement 2.** Impact of signal-to-noise on identification of the edge map.
**Figure supplement 2—source data 1.** Peak Signal to Noise Ratio (PSNR) values of images with increasing noise added.

percentage of thresholded area of the whole original image. The E-cadherin mask is also used to subtract an ROI from an image of a distinct marker (i.e. F-actin), so that mostly the signal localized at

contacts is considered. Junction Mapper implements novel quantification tools (corners, length and area) and a variety of novel primary and secondary parameters expressed per individual junction.

The main advantage of Junction Mapper is to efficiently segment junctions in a variety of cell types and of different junction patterns, from linear to fragmented or disrupted contacts. The software detects the contacting interface between two cells and obtains a skeletonized edge map (summarised stepwise in *Figure 2—figure supplement 1*; see Materials and methods). Although semi-automated, users can adjust the outline manually by removing incorrect or adding missing lines, with subsequent refinement of the line location and geometry by the software algorithm (detailed in Appendix 1). A dilation step is applied (user controlled) to define the area to be quantified around the skeletonized map (*Figure 2—figure supplement 1*). To identify individual junctions for further quantification, the software then automatically identifies each cell–cell corner (i.e. point of contact between three or more cells, see its mathematical definition in Appendix 1). The number and location of corners in each cell can also be manually adjusted by the user (Appendix 1).

Images obtained at different resolutions can be used for analyses in Junction Mapper (*Supplementary file 1*). However, the highest quality images possible should be used, as resolution may impact the ability to detect individual clusters of the junction marker. For defining the automated skeletonized edge of each image, a suitable signal-to-noise ratio is necessary to ensure enough contrast to differentiate staining at junctions from the cytoplasm (*Figure 2—figure supplement 2*). The higher the Peak Signal to Noise ratio, the easier it is for the program to automatically delineate the cell outline skeleton. For efficient skeleton definition the ratio should be above 22 dB (*Figure 2—figure supplement 2*). Using the automated skeleton as a start point, the user can refine the cell outline map by manually drawing or removing lines to close any gaps or correct deviations, particularly in cells with reduced staining at cell–cell contacts (*Figure 2—figure supplement 1 to 2*). The higher the fragmentation of junctions in the image, the more user input is necessary to define the final skeleton outline for quantification.

User-controlled threshold selection is done by inspection of the image with a slider component on the software interface (Appendix 1). Thresholding aims to reduce background without removing pixels from contact areas. The skeleton obtained is then projected onto the original thresholded image to segment the area of interest and proceed with quantification of each junction for up to two different junctional markers per image (see below). The measurement of the different parameters is performed simultaneously and automatically, in an unbiased manner (*Figure 2*). The output per junction is produced as an Excel file that contains the selected image, skeleton and the quantification of pre-designed measurements and parameters (see below).

## Primary parameters and validation

We envisage that different phenotypes of junction perturbation may require a distinct set of parameters to appropriately capture the disruption features (*Figure 1*). Towards this goal, we propose a variety of parameters to be used as *bona fide* readouts and measure information that is based on intensity, area and length of the interface and cell–cell contact (*Figure 3A*; Appendix 2). The following concepts are defined and used herein. Interface is the contacting membrane between two neighbouring cells and delimited by cell–cell corners. A cell–cell contact or junction is the region of the interface covered by adhesion receptors (e.g. cadherin staining). Junctions may or may not extend to the whole of the interface (corner-to-corner), and can appear fragmented or dotted (*Figure 1*, *Figure 3A*). Area is the dilated region around the skeleton and is set to encompass the width of a junctional marker staining (which can vary in thickness). Contour is the length measurement of the outline of the skeleton between defined points (interface or junction). Finally, the straight-line length is defined as the shortest distance (Euclidian distance) between corners that form the boundaries of one junction or interface.

Primary parameters obtain the basic metrics (area, contour and straight-line length) of each interface and junction selected in the monolayer (*Figure 3A*, Appendix 2). A key innovation of the Junction Mapper is its ability to quantify junction marker staining that is not contiguous and that does not extend to both corners. New parameters assess area or contour occupied by fragments of the junctional staining (Fragmented Junction Contour and Junction Area). To validate the length-based measurements, individual contacts disrupted by H-Ras expression were quantified and, as predicted, decreasing values for the contour of the interface, junction or fragmented junction were revealed for each assessed junction (*Figure 3B–C*). Although measurements of contacts of control cells also

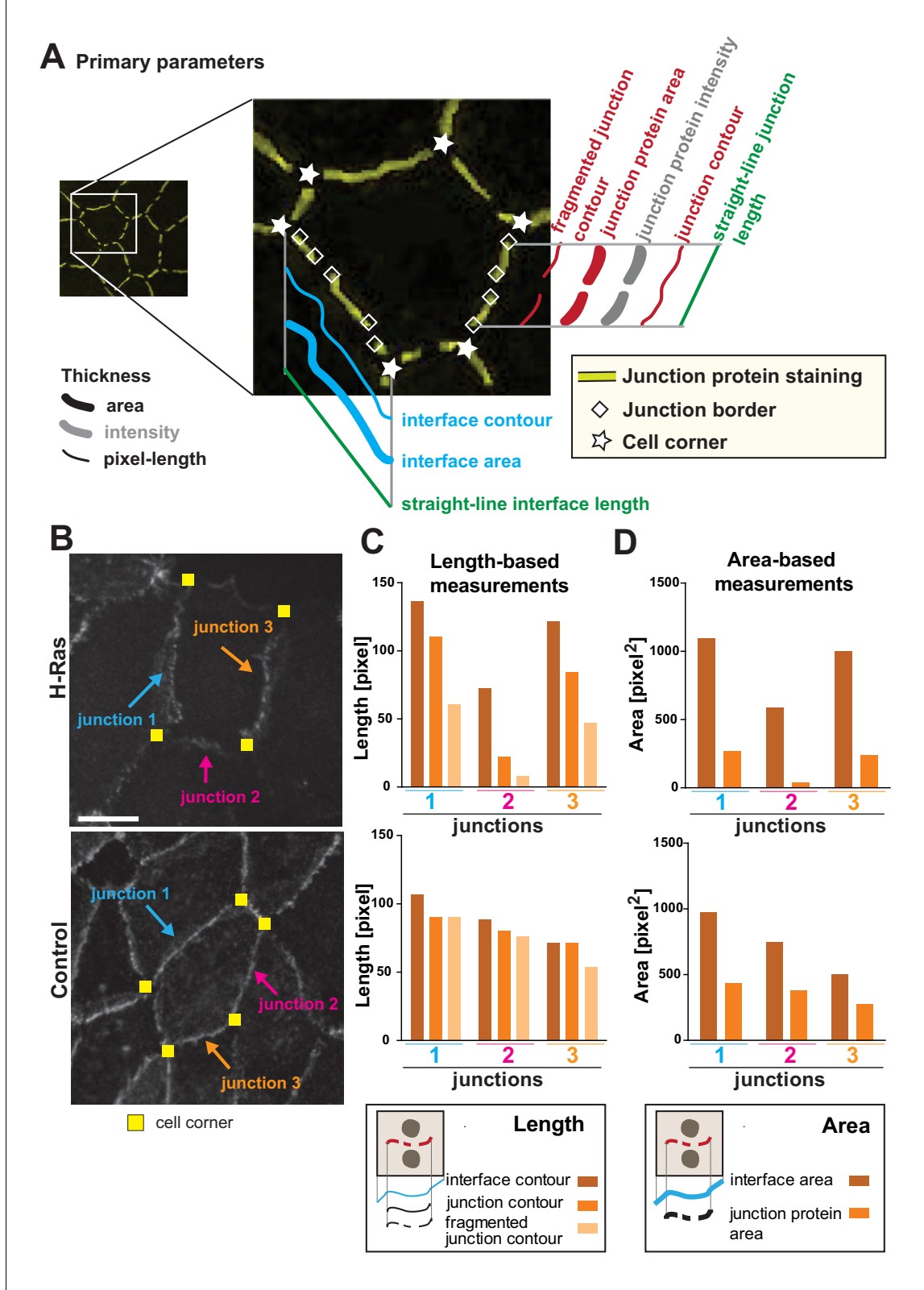

**Figure 3.** Junction Mapper – parameters. (A) Diagram highlighting the concepts that underpin the primary parameters measured by the software. An image of a cell with a hypothetical disruption of E-cadherin at cell–cell contacts is shown: stars mark the corners delimiting each contacting interface, and rhombus shapes mark the edges of each fragment of cadherin staining that we define as junctions. In each cell, measurements are made to assess the properties of each interface (corner-to-corner) and each junction (may be adjoining or disrupted in multiple fragments): the hypothetical length

*Figure 3 continued on next page*

*Figure 3 continued*

(straight line interface length or Euclidian distance between two points), the contour (connection of brightest pixels along the curvature of the staining/interface) and area (defined dilated region around cell–cell borders). Intensity is also measured within the dilated area. (**B-D**) The primary parameters were validated to assess measurements were as predicted from their definition. Selected junctions from cells expressing activated H-Ras or controls (**B**) were quantified for length-based measurements (**C**, contours of interface, junction or fragmented junctions) or area-based measurements (**D**, areas of interface or junction). Diagrams at the bottom of graphs in C and D show the measurements performed. Coloured arrows point to selected junctions quantified. Each contacting interface is delimited by corners visualized by yellow squares. Scale bar = 10 µm. Detailed steps are described in Appendix 2.

The online version of this article includes the following source data and figure supplement(s) for figure 3:

**Source data 1.** Data used to validate length- and area-based primary parameters.
**Figure supplement 1.** The impact of user-controlled dilation step settings on measurements.
**Figure supplement 1—source data 1.** Data to validate the impact of user-controlled dilation settings.
**Figure supplement 2.** The impact of user-controlled thresholding settings on measurements.
**Figure supplement 2—source data 1.** Data to validate the impact of user-controlled thresholding settings.

decline between these three parameters, they did not decline to the same extent as the ones of junctions from expressing cells (*Figure 3C*). The area-based measurements Interface Area and Junction Area of the same junctions followed similar pattern (*Figure 3D*). We envisage that the primary parameters are useful to show extension or retraction of the contacting interface/junctions, increased fragmentation of the staining of junction marker or fluctuations of global staining intensity at cell–cell contacts.

We next assessed the impact of user-defined settings (dilation and threshold) on the primary parameters. The variable dilation settings define areas for quantification and permits the user to account for different thickness of the staining at cell–cell contacts, wavy or undulated junctions. Increasing the dilated area did not affect the length of contacting interface or fragmented junction (*Figure 3—figure supplement 1A–C*), but positively correlated with E-cadherin intensity and area (*Figure 3—figure supplement 1D*). Similarly, larger dilation values better captured the amount of VE-cadherin present in endothelial junctions as they acquire a zig-zag conformation after thrombin stimulation (*Figure 3—figure supplement 1E–F,H*). A trade-off is necessary between the amount of dilation and thresholding, so that the contribution of cytoplasmic staining is minimized with larger dilation values.

The effect of thresholding was tested using the same images (*Figure 3—figure supplement 2A–B,E–F*). Of note is that the precise outline of VE-cadherin zig-zag staining was not recognized by the edge map produced (red line, *Figure 3—figure supplement 2E–F*). Increasing the threshold applied to the original images decreased the measured cadherin area and intensity (*Figure 3—figure supplement 2D, H*) and the length of junction fragments was severely reduced (*Figure 3—figure supplement 2C,G*; see definition in Appendix 2). The interface contour values were not affected by thresholding, as the interface is delimited by corners and independent of intensity values (*Figure 3—figure supplement 2C,G*). Taken together, we concluded that the primary parameters measure the length and area of various features as predicted and are selectively modulated by user-controlled settings in accordance to their definition (Appendix 1).

## Secondary parameters

The availability of a repertoire of parameters is a unique advantage of the software for mapping distinct phenotypes. Secondary parameters (*Scheme 1*, Appendix 2) are derived from the primary metrics above and aim to normalise the measurements to the size of each junction or contacting interface (length or area). The Linearity Index measures how straight an interface between two cells is, as proposed by Takeichi and colleagues (*Otani et al., 2006*). Coverage Index is a length measurement of the percentage of the interface length that is covered by the junction marker staining and has been previously used manually in the lab (*Lozano et al., 2008*). Three different parameters quantify the distribution of a junctional marker along the interface. First, Interface Occupancy, measures the area occupied by the marker within the Interface Area of each junction. Second, Junction Protein Intensity per Interface area calculates the fluorescence intensity within the Interface Area. Finally, Cluster Density is the junction protein intensity level within the area delimited by its staining (i.e. Fragmented Junction Area, which considers any fragmentation of the staining; additional parameters are described in Appendix 2). We predict that the secondary parameters are more useful to

compare the accumulation or removal of specific markers, their density and relative changes when comparing across different samples.

The Coverage Index parameter calculated by Junction Mapper was validated by manual quantification using FIJI on cells expressing active Rac1 (*Supplementary file 2A-B*). The manual quantification used straight-line measurements (*Lozano et al., 2008*), rather than the more precise contour measurement by Junction Mapper (*Scheme 1*). Using either quantification, manual or Junction Mapper, a significant statistical difference was observed between values of control and Rac1-expressing cells (*Supplementary file 2C*). Lower values were obtained with the manual quantification method as predicted when using straight-lines for measurement. However, when controls values were compared between the two methods, no statistical difference was obtained (*Supplementary file 2C*).

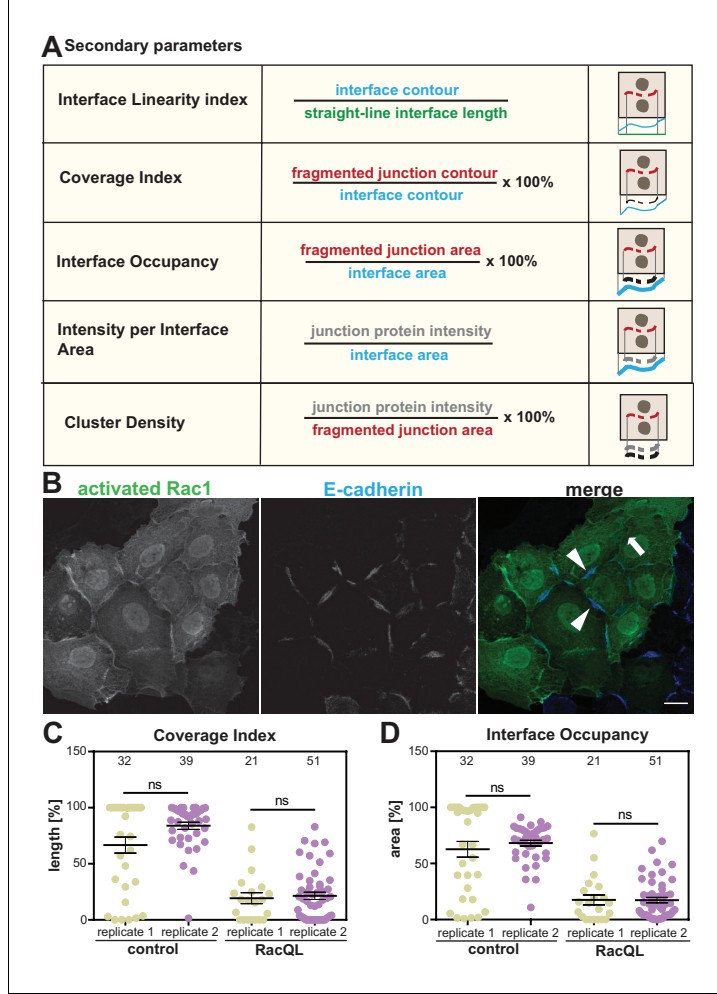

**Scheme 1.** Secondary parameters. (**A**) Novel parameters were defined to normalise the quantifications with respect to the area or length of contacts. The secondary parameters assess the configuration of the contacting interface (Interface Linearity Index), how much the staining of a junction marker occupies the interface length (Coverage Index) or area (Interface Occupancy). The distribution of junction marker is measured in two ways: their intensity levels within the area occupied by the junction fragments (Cluster Density) or the contacting interface (E-cadherin intensity at interface area). Detailed information of the calculation of parameters is described in Appendix 2. (**B-D**) Reproducibility of quantification by Junction Mapper in independent biological replicates. (**B**) Keratinocytes expressing activated Rac1 (green) or controls (non-expressing cells) were stained for E-cadherin (blue). (**C-D**) Images obtained from two independent biological replicates (replicate 1 and replicate 2) were processed to obtain the secondary parameters Coverage Index (C) or Interface Occupancy (D). Numbers at the top inside graphs show the number of junctions quantified in each sample from two biological replicates (N = 2); ns, non-significant. Arrowheads point to residual E-cadherin staining; thick arrow shows lack of cadherin staining. Scale bar = 20 μM.

Thus, the parameters give results as expected from previous manual methods. In addition, the reproducibility of secondary parameters was assessed in independent biological replicates by the same user. Across biological replicates of cells overexpressing active Rac1, the absolute values obtained for each sample were slightly different (*Scheme 1B-D*). However, the overall result is the same between replicates: a reduction of Coverage Index and Interface Occupancy of E-cadherin at junctions from Rac1 expressing cells (*Scheme 1B-D*). Comparison of control values between biological replicates (or between Rac1 expressing cells) was not statistically different.

## User-bias validation

The robustness of Junction Mapper with respect to user bias was tested by: (i) defining the skeleton and corners and (ii) quantifying the same samples by different users. First, the quality control of skeleton and corners is subjective (*Supplementary file 3A-B*). Confluent epithelial monolayers with clearly defined junctions and corners produced less output variability from different users than images with disrupted contacts (*Supplementary file 3A-D*). We found that a particular hot spot for differences was the identification of cell corners in disrupted contacts, where corners are often not covered by cadherin staining (*Supplementary file 3B*). Second, pair-wise comparison of primary parameters of matched epithelial junctions obtained independently by two users showed significant differences (*Supplementary file 3C-D*).

However, when secondary parameters were calculated, the profiles obtained by the two users were similar, aside the absolute values being different (*Supplementary file 3E-F*, see also *Figure 7—figure supplement 1*). This can be explained by the fact that the user-dependent variability partially auto-correct itself, as values are normalised to the interface and junction area or length which are non-reciprocally impacted by user bias. For example, a larger dilation setting will generate a larger junction area containing a higher number of E-cadherin pixels; when cluster density is calculated (E-cadherin intensity/fragmented junction area), the ratio will not differ extensively between different users.

We conclude that, in addition to the impact of dilation and thresholding settings, Junction Mapper results are impacted by user influence on the cell edge map. This influence is stronger in cell images with disrupted cell–cell contacts, where higher frequency of inaccurate corner and skeleton detection occurs. Hence more manual edge correction is required the more robust secondary parameter measurements should be considered. Furthermore, absolute comparisons can not be made between experiments. Instead, normalization of secondary parameters to controls in each replicate will facilitate comparisons.

## Distinct oncogenes trigger different patterns of junction disruption in epithelia

We addressed whether Junction Mapper parameters could identify distinct features and patterns of junction disruption by different stimuli. We tested images of epithelial cells transfected with oncogenic Rac (H-Ras$^{G12V}$; *Figure 4A*), constitutively activated Rac1 (Rac$^{Q61L}$) or activated Src (Src$^{Y527F}$; *Figure 4—figure supplement 1*). Previous visual analysis indicated that perturbation of cell–cell contacts occurs more efficiently between two Rac$^{Q61L}$-expressing neighbouring keratinocytes (*Braga et al., 1997*; *Lozano et al., 2008*). To validate this quantitatively, junctions were classified into two groups (i) between two expressing cells (ee) and (ii) between one expressing and one non-expressing cell (en). Control junctions from the same image were quantified from cells without oncogenic H-Ras expression. Measurements of the primary parameters (*Figure 3*) confirmed that junction contour and area were more severely disrupted when two neighbouring cells contained exogenous H-Ras$^{G12V}$ (*Figure 4D–E*). The interface of mosaic contacts shared by expressing and non-expressing cells (en) were not significantly different from controls (*Figure 4B,C,F*).

The interface between cells containing oncogenic H-Ras (ee) was significantly longer and larger than control cells (Interface Contour and Interface Area, *Figure 4B,C*). Irrespective of the longer contacting interface, the contour and the area of E-cadherin stained fragments were considerably altered upon expression of H-Ras$^{G12V}$ (Fragmented Junction Contour and E-cadherin Area, *Figure 4D,E*). Finally, the length between cell–cell corners was significantly longer between controls and two adjacent transfected cells (Straight-line Interface Length, *Figure 4F*). Based on the

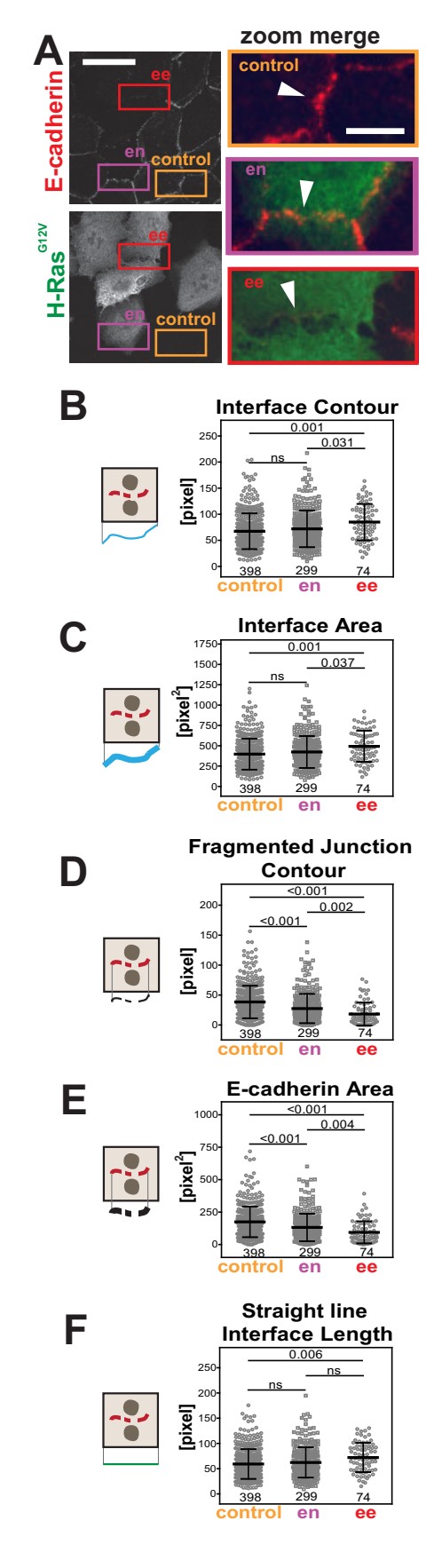

quantification of the primary parameters, oncogenic H-Ras expression induces a longer contacting interface between cells and a progressive fragmentation of E-cadherin staining.

The primary parameter measurements following expression of activated Rac1$^{Q61L}$ or Src$^{Y527F}$ (**Figure 4—figure supplement 1**) showed distinct alteration profiles when compared between each other and to activated H-Ras (**Figure 4**). For example, active Src expression did not promote elongation of the contacting interface or an increase in interface area (**Figure 4—figure supplement 1H–I**), while active Rac1 did not induce fragmentation of cadherin staining (**Figure 4—figure supplement 1D–E**). The above data may indicate that distinct subsets of parameters can differentiate alterations by different stimuli, thereby providing a unique disruption profile (**Figure 5**). While data are from one technical replicate only, the high number of junctions quantified per sample is still sufficient to indicate significant differences among groups (**Supplementary file 1**). However, further experimentation is required to confirm the patterns observed.

The secondary parameters (**Scheme 1**) were designed so that the distribution of a junctional marker is normalised to the area or length of the interface or junction between neighbouring cells. Although Intensity is a primary parameter, it was also included here for comparison with other published studies. The patterns of the secondary parameters Interface Occupancy and Cadherin Intensity at Interface Area essentially followed the corresponding primary parameters E-cadherin Area and E-Cadherin Intensity (**Figures 4** and **5**). Yet, the differences between groups are more apparent in the secondary parameters: data are less scattered with fewer outliers when compared to primary parameters (**Figure 5**).

We decided to focus on junctions that were shared by two transfected cells (ee), where phenotypes are clearer (**Frasa et al., 2010**; **Lozano et al., 2008**). Upon transfection of activated forms of H-Ras, Src or Rac1, a progressive disappearance of E-cadherin from the interface between cells was observed in different patterns (**Figure 5A,C,E**). When compared to controls, the undulation of the interface was increased among cells expressing H-Ras$^{G12V}$ or Src$^{Y527F}$ (higher Interface Linearity Index, **Figure 5B,D**), but remained unchanged for Rac1$^{Q61L}$-perturbed junctions (ee, **Figure 5F**). These data indicate that, as E-cadherin is removed from junctions, the interface between H-Ras$^{G12V}$ and Src$^{Y527F}$ expressing cells becomes less tensile. The

**Figure 4.** Primary parameters quantification of H-Ras-dependent junction perturbation. (**A**) Human normal keratinocytes were transfected with pRK5-myc-H-Ras$^{G12V}$, fixed and stained with anti-E-cadherin and anti-myc antibodies. Images are shown of E-cadherin staining and myc staining as a marker of transfected cells. Coloured rectangles mark areas shown as a zoom on the left of the images and highlight control junctions (orange), junctions between expressing and non-expressing cells (en, purple) or between two transfected cells (ee, red). Arrowheads point to E-cadherin staining. (**B-F**) Quantification of the primary parameters using Junction Mapper. Graphs are plotted showing values of each parameter (Y axis) versus different junction types (X-axis). The parameter name is at the top of each graph and a diagram representing the quantification is shown on the left of its corresponding graph. Data is from one experiment (technical replicate) and the number of junctions analysed for each condition is found at the bottom of the graphs, below each scatter box plot. Statistical analysis was performed using One-way ANOVA, followed by Games-Howell post-hoc test. Non-significant (ns) and significant p-values (<0.05) are placed inside graphs. Scale bar = 20 µM or 10 µM (zoom images).

The online version of this article includes the following source data and figure supplement(s) for figure 4:

**Source data 1.** Primary parameters data to prepare *Figure 4B-F*.

**Figure supplement 1.** Primary parameters quantification of junction perturbation by activated Rac1 or activated Src.

**Figure supplement 1—source data 1.** Primary parameter data for activated Rac1 and Src.

percentage of the interface area occupied by E-cadherin receptors was decreased in all samples (Interface Occupancy, *Figure 5B,F*), but did not reach significance in Src-expressing cells (*Figure 5D*).

The intensity levels of cadherin at junctions was significantly reduced following transfection of activated H-Ras or Src when measured as raw values (E-cadherin Intensity) or corrected per contacting area between two cells (E-cadherin Intensity per Interface Area) (*Figure 5B,D*). In contrast, after active Rac1 expression, neither parameter was significantly changed. Consistent with the distinct phenotype of junction perturbation seen in Rac1$^{Q61L}$-transfected cells, the density of cadherin clusters was decreased in H-Ras$^{G12V}$ and Src$^{Y527F}$, but slightly augmented in Rac$^{Q61L}$.

These data are summarized as a diagram in *Scheme 2*. We concluded that Junction Mapper quantification can capture the various phenotypes of junction perturbation and collectively define a specific profile for each oncogene. Our preliminary observations suggest that activated Rac1 does not significantly reduce the overall levels of E-cadherin at junctions, since receptor intensity at the interface area is not reduced. Rather, Rac1 activation progressively redistributes receptors at the junction (reduced Interface Occupancy and higher Cluster Density), maintaining straight, linear junctions. In contrast, activated H-Ras or Src disrupt junctions via fragmentation and removal of E-cadherin consistently throughout the contacting interface (decreased in Interface Occupancy and Cluster Density), with concomitant undulation of cell–cell contacts (i.e. reduced tension or stiffness at junctions).

## Dynamic range of the measurements of cell–cell contact phenotypes

Disassembly of junctions by oncogenes is a severe phenotype, often leading to complete dissolution of contacts. However, other stimuli (i.e. differentiation, protein depletion, growth factor or drug treatment) may induce a milder phenotype that is not easily quantified. We asked whether the Junction Mapper could efficiently detect small changes in E-cadherin levels or distribution under different conditions (*Figure 6*). Datasets were obtained where the role of actin-regulatory proteins in junction formation was investigated in normal keratinocytes (*Erasmus et al., 2016*). Depletion of CIP4 (a regulator of cadherin trafficking) (*Leibfried et al., 2008*; *Rolland et al., 2014*), VAV2 (an exchange factor for Rho, Rac2 and Cdc42) (*Vigil et al., 2010*) or EEF1A (an actin bundling protein) (*Mateyak and Kinzy, 2010*) results in modest, but significant fluctuations in E-cadherin at junctions (20%–30%) using a thresholding method (*Erasmus et al., 2016*).

When analysed with Junction Mapper, CIP4, VAV2 or EEF1A siRNA did not interfere with the linearity of the contacting interface (*Figure 6B,D,F*), in line with the appearance of normal, linear junctions. Consistent with our previous findings (*Erasmus et al., 2016*), a small decrease in E-cadherin intensity was observed in both CIP4- and VAV2-depleted cells (*Figure 6B,F*), while EEF1A siRNA promoted the unusual phenotype of higher levels of cadherin receptors (*Figure 6D*). These distinct patterns were also seen when E-cadherin intensity and area were normalized to the interface area (Intensity per Interface Area and Interface Occupancy, respectively). Strikingly, despite the similar

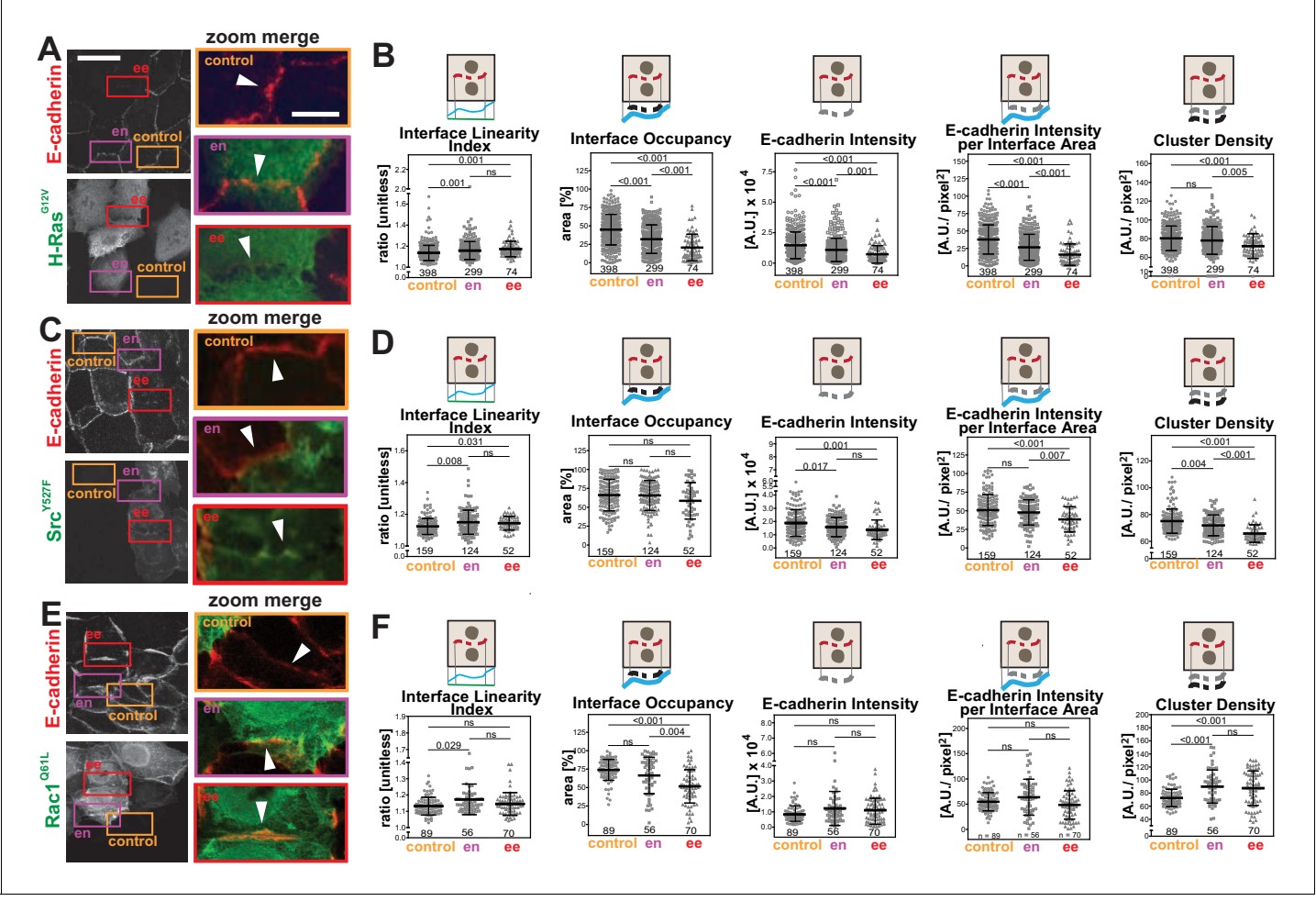

**Figure 5.** Distinct profiles of junction disruption by oncogenes. Human normal keratinocytes were transfected with pRK5-myc-H-Ras[G12V] (A), pEGFP-Src[Y527F] (C) or pRK5-myc-Rac1[Q61L] (E). Cells were fixed and stained with anti-E-cadherin and, for (A) and (E), anti-myc antibodies. Images are shown of E-cadherin and transfected cells (anti-myc or GFP). Coloured rectangles mark areas shown as a zoom on the left of the images and highlight control junctions (orange), junctions between expressing and non-expressing cells (en, purple) or between two transfected cells (ee, red). Arrowheads point to E-cadherin staining. (B, D, F) Quantification of different parameters obtained with Junction Mapper. Graphs are plotted to show values of each parameter (Y-axis) versus different junction types (X-axis) for H-Ras[G12V] (B), Src[Y527F] (D) and Rac1[Q61L] (F). The parameter name and a diagram representing the quantification are shown on top of each graph. Technical (H-Ras, Rac1) or biological replicates (Src, N = 2) were analysed. Number of junctions quantified in each condition is shown at the bottom of the graphs, below each scatter box plot. Statistical analysis was performed using One-way ANOVA, followed by Games-Howell post-hoc test. Non-significant (ns) and significant p-values ($<0.05$) are placed inside graphs. Scale bar = 20 µM or 10 µM (zoom images).

The online version of this article includes the following source data for figure 5:

**Source data 1.** Secondary parameter data on oncogenic junction disruption used for *Figure 5* graphs.

reduction in E-cadherin intensity levels following VAV2 and CIP4 depletion, receptors were removed in different ways from junctions. The clusters of cadherin receptors were less dense with lower levels of CIP4, while upon VAV2 siRNA, the density of the clusters slightly increased (Cluster density, *Figure 6B,F*).

Thus, the discrete changes in junctions result from reduced cadherin levels throughout the contacting interface by lower density of receptor clusters (CIP4 RNAi) or localised E-cadherin removal and redistribution into denser clusters (VAV2 RNAi; *Figure 6B,F*). In contrast, EEF1A depletion augmented E-cadherin Intensity and Cluster Density along the interface. Taken together, these data highlight the ability of the Junction Mapper to detect phenotypes reproducibly and with good dynamic range. *Scheme 3* summarizes the changes and distinct profiles detected by Junction Mapper in our validation experiments.

## Endothelial junctions and cardiomyocyte intercalated discs

We next asked whether the software would be applicable to endothelial cells and cardiomyocytes that can have junctions differently shaped when compared with epithelial contacts. Both types of junctions are considerably more fragmented than epithelial contacts, and thus it was not clear whether the parameters quantified by the Junction Mapper would be suitable.

Thrombin is a serine protease that stimulates protease-activated receptor (PAR) in endothelial cells to increase vascular permeability in inflammation and injury (*Figure 7A*) (*Kumar et al., 2009*). Visually, the junctions of endothelial cells treated with thrombin are quite distinct (*Malinova and Huveneers, 2018*), but so far it has not been possible to evaluate differences quantitatively. There were no significant changes in the Interface Linearity Index (*Figure 7A,B*), consistent with the limitation of Junction Mapper to skeletonize zig-zagged junctions and thus measure their length (*Figure 3—figure supplement 2E–F*).

However, with appropriate dilation values (*Supplementary file 1*, *Figure 3—figure supplement 1*), alterations were detected for area-based measurements. In thrombin-treated cells, VE-cadherin distribution along the interface area (Interface Occupancy), intensity of VE-cadherin staining and normalised intensity per contacting interface were substantially higher (*Figure 7C–E*). The density of VE-cadherin clusters was also enhanced upon treatment with thrombin (*Figure 7F*), implying that a higher number of receptors is recruited per contact area. Of note is that the analysis of stimulated endothelial cells, with their typical junction morphology and gaps, was strongly influenced by the user-controlled settings (*Figure 7—figure supplement 1A–C*). Yet, although raw values of each junction varied with different user settings, the overall trend and conclusion remained the same (*Figure 7—figure supplement 1D–F*), consistent with our previous comparative analyses (*Supplementary files 2–3*).

Rat neonatal cardiomyocytes were treated with phenylephrine (PE) as a model to induce hypertrophy (*Miragoli et al., 2011*; *Simpson, 1985*). Cells were co-stained with β-catenin as a junctional marker and connexin 43 (*Figure 8A*), a protein found in gap junctions, a structure necessary for synchronization of cardiomyocyte beating. The appearance of both markers at junctions was considerably fragmented (dotted appearance; *Figure 8A*), suggesting that parameters that consider the staining area would be the most appropriate.

| Parameters | H-Ras^G12V | | Src^Y527F | | Rac1^Q61L | |
|---|---|---|---|---|---|---|
| | control | ee | control | ee | control | ee |
| Interface Linearity Index | ● | ● | ● | ● | ● | ● *ns |
| Interface Occupancy | ● | ● | ● | ● *ns | ● | ● |
| E-cadherin Intensity | ● | ● | ● | ● | ● | ● *ns |
| E-cadherin Intensity/ Interface Area | ● | ● | ● | ● | ● | ● *ns |
| Cluster Density | ● | ● | ● | ● | ● | ● |

**Scheme 2.** Overview of different profiles of junction disruption caused by expression of oncogenic Ras, Src or Rac1. Different parameters are normalised to controls (junctions from non-expressing cells) arbitrarily set as 100 (orange colour). Values are represented as circles of proportional sizes for junctions between two expressing cells (red colour). Non-significant values are shown in pink colour (ns).

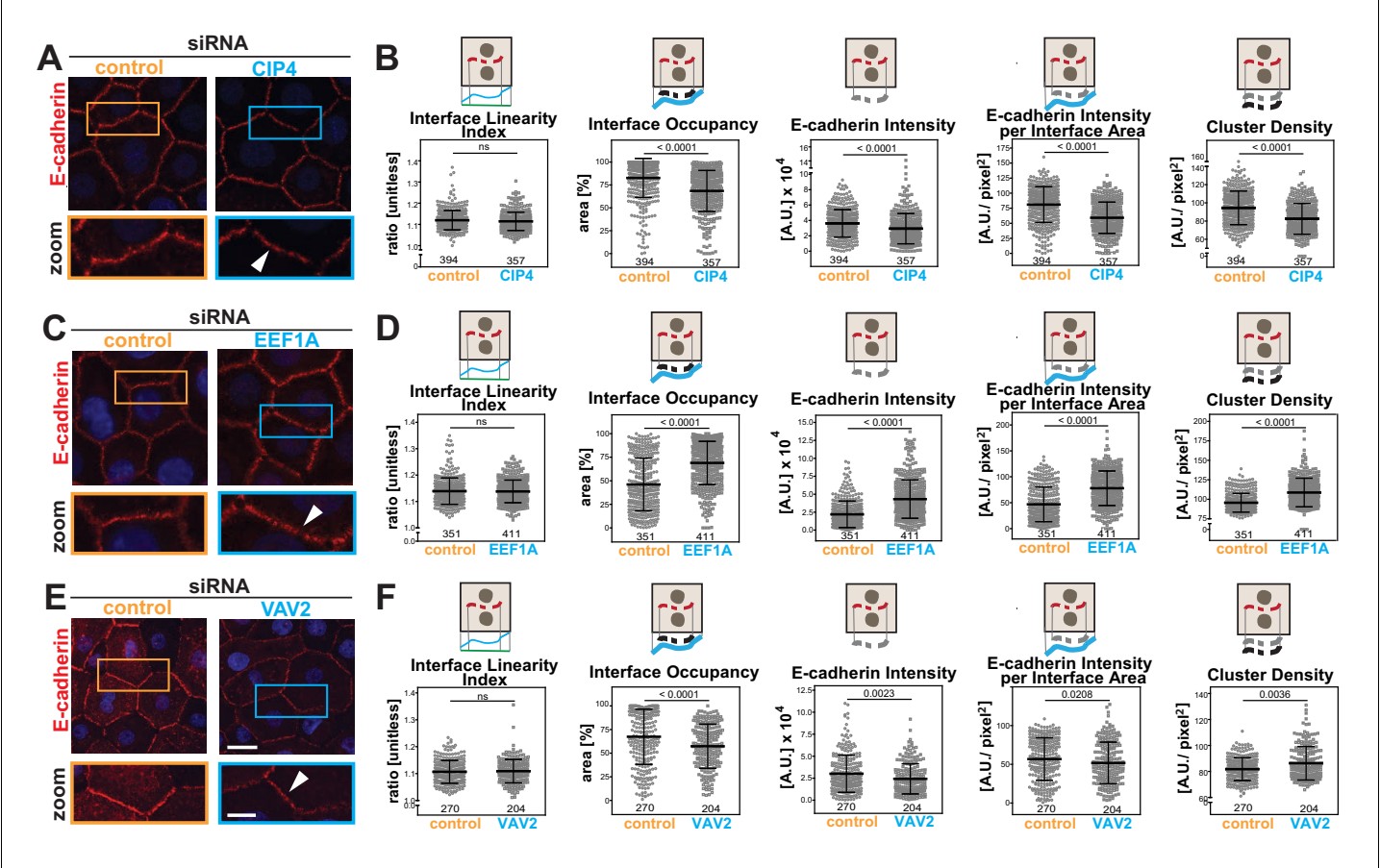

**Figure 6.** Junction Mapper reliably quantifies mild phenotypes. (A) Human normal keratinocytes were depleted of different cytoskeleton-associated proteins using siRNA against CIP4 (A), EEF1A (C) or VAV2 (E). Cells were fixed and stained for E-cadherin, and images acquired for control (non-targeting oligos) and depleted cells. Areas marked by coloured rectangles are shown as a zoom underneath the images. Arrowheads point to E-cadherin staining. (B, D, F) Quantification of different parameters obtained with Junction Mapper. Graphs are plotted with values on the Y-axis and control and siRNA samples on the X-axis for CIP4 (B), EEF1A (D) and VAV2 (F). The parameter name and a diagram representing the quantification are shown on top of each graph. Junctions analysed in each condition were obtained from technical replicates and numbers assessed is shown at the bottom of the graphs, below each scatter box plots. Statistical analysis was performed using Mann-Whitney U-test. Non-significant (ns) and significant p-values (<0.05) are placed inside graphs. Scale bar = 20 µM or 10 µM (zoom images).

The online version of this article includes the following source data for figure 6:

**Source data 1.** Data from siRNA experiments used in graphs in *Figure 6B, D and F*.

The software used the skeleton and dilated area of β-catenin as a mask as described by *Erasmus et al. (2016)* to segment the area where connexin 43 was localised. The Interface Linearity Index of β-catenin (*Figure 8B*) or connexin 43 (*Figure 8C*) was not significantly altered upon induction of hypertrophy *in cellulo*. Instead, hypertrophy stimulation promoted higher Interface Occupancy and higher levels of β-catenin and connexin 43 at junctions (raw intensity or corrected by the interface area; *Figure 8B,C*). However, the density of β-catenin clusters was significantly reduced, while connexin 43 density was augmented in PE-treated cells, in a small but significant manner. Thus, the two markers are modulated differently by hypertrophy stimulation *in cellulo*. Increased levels of β-catenin are achieved by less dense clusters spread along the interface, while connexin 43 molecules are clustered more densely, suggesting localised stimulation of gap junction formation. These data demonstrate the use of Junction Mapper for multiple cell types and its power to correlate the distribution of different proteins at cell–cell junctions. The data from cardiomyocytes and endothelial cells are summarized in *Scheme 4*. A heuristic approach on how to define the user-dependent settings and the parameters to use for these cell types are described in *Supplementary file 4*.

# Discussion

Junction configuration and adhesion receptor organization at contact sites are maintained via a complex interplay of distinct cytoskeletal filaments and associated regulatory proteins, with an exquisite regulation by diverse pathways and cellular contractility. At cell–cell contact sites, the challenge remains to translate the regulation of junctional components into functional and appropriately shaped cell–cell contacts. Junction Mapper facilitates the profiling of cell–cell contact behaviour with a variety of novel parameters in a fast, robust and reliable manner. Collectively, the parameter repertoire indicates how effective cell–cell adhesion is, identifies altered patterns of receptor distribution and guides experimental design to unravel the underlying molecular mechanisms.

Junction Mapper provides a semi-automated computer vision solution with broad applicability. The most innovative aspects of Junction Mapper are, first, the measurement of receptor density and occupancy, via the normalization of the junction marker intensity, length and area to the available contacting interface and cell–cell contact area. Second, the automatic quantification of fragmented staining at a junction, which has not been feasible previously and, third, the correlation of the parameters from two junctional markers along the same junction. Underpinning the above aspects is the ability of Junction Mapper to detect cell borders and corners when considerable disruption is observed. Machine learning algorithms certainly have the potential to improve and automate segmentation success, especially of highly fragmented contacts. However, the availability of training image datasets with boundaries manually-generated is still a bottleneck (*Häring et al., 2018*). One possible use of the software is therefore the generation of precise cell outline skeletons that can be used to train machine-learning algorithms in the future.

While semi-automation has been implemented during the image processing by Junction Mapper, user contribution is necessary and essential to perform the quality control of skeleton outline, corner positioning and setting up the dilation and threshold levels in a given dataset. User bias is particularly relevant in images with very disrupted and irregular junctions, and absolute values are not suitable for comparison across experiments in some cases. However, user bias can be minimized. Analysis of biological replicates or independent analysis by different users show that (i) the secondary parameters are more robust against user bias and (ii) obtained results are similar across replicates when comparing control and treated samples.

**Scheme 3.** Overview of different profiles of junction disruption following depletion of CIP4, EEF1A or VAV2. Different parameters are normalised to controls (junctions from cells treated with non-targeting oligos) arbitrarily set as 100 (orange colour). Values are represented as circles of proportional sizes for targeting siRNA-treated samples (blue colour) relative to controls. Non-significant values are shown in light blue colour (ns).

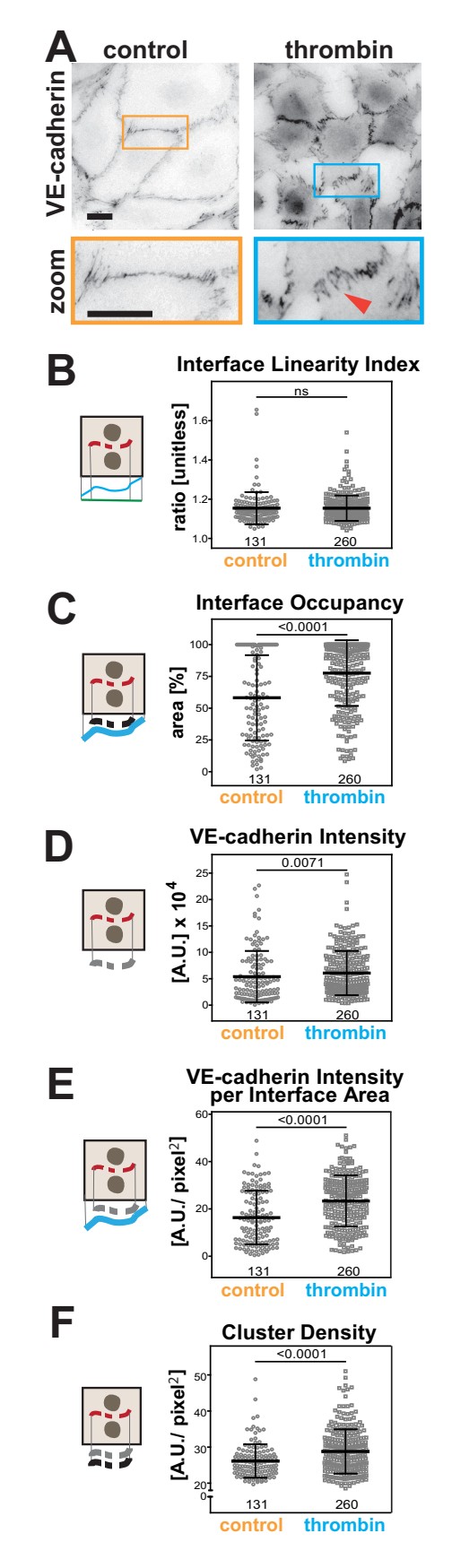

**A** control thrombin

VE-cadherin

zoom

**B** Interface Linearity Index

**C** Interface Occupancy

**D** VE-cadherin Intensity

**E** VE-cadherin Intensity per Interface Area

**F** Cluster Density

The software can detect both major and minor changes at junctions in different experiments. Unexpected and distinct profiles emerge from junction disruption by oncogenes that potently remove E-cadherin from contacts. The junctional defects caused by H-Ras$^{G12V}$ or Src$^{Y527F}$ illustrate the novel parameters that measure junction fragmentation and the specific density of E-cadherin in remaining fragments. In contrast, the perturbation profile induced by expression of Rac1$^{Q61L}$ is not appropriately quantified by intensity alone and may be better assessed by 'Interface Occupancy', 'Coverage Index' and 'Cluster Density'. Among the repertoire of pre-defined parameters, we find that a subset is highly likely to assess the features of a particular junction phenotype. Phenotypes not yet analysed could present additional possibilities to improve the Junction Mapper repertoire in future studies.

Junction Mapper analysis confirms mild phenotypes previously observed with manual, threshold-based quantifications of whole images (*Erasmus et al., 2016*), that indicate either an increase (EEF1A siRNA) or a decrease (CIP4 or VAV2 siRNA) in the intensity levels of E-cadherin. In addition, new Junction Mapper parameters uncover distinct features, that is that EEF1A depleted cells have higher E-cadherin occupancy and augmented cluster density when compared to controls. The profiling with new parameters underscores the potential of Junction Mapper to differentiate among distinct, subtle modes of junction perturbation. Yet, the conceptual significance of such alterations remains to be consolidated in future experiments and with additional biological replicates.

The plasticity of endothelial junctions is well established (*Radeva and Waschke, 2018*), but their unique responses to different stimuli have been challenging to quantify (*Häring et al., 2018*). The remarkable zig-zag pattern of thrombin-stimulated endothelial junctions correlates with increased vascular permeability (*Malinova and Huveneers, 2018*) but it is not recognized by the automated skeleton definition. Using area-based parameters, we find that the contacting interface is occupied more efficiently, with higher density of VE-cadherin receptors at endothelial junctions upon thrombin treatment. Thus, it seems that the increased levels of VE-cadherin at junctions may promote stronger endothelial adhesion, which is relevant to sustain elevated intracellular tension and contractility induced by thrombin stimulation. These results are consistent with the role of mechanical

**Figure 7.** Quantification of endothelial junction alterations triggered by thrombin stimulation. (A) HUVEC were treated with thrombin for 10 min, fixed and stained for VE-cadherin as a marker of endothelial contacts. Inverted images and a zoom are shown. Arrowheads show VE-cadherin staining. (B-F) Quantification of different parameters obtained with Junction Mapper. Graphs are plotted with values on the Y-axis and samples (control or thrombin-treated) on the X-axis. The parameter name is shown on top of each graph and a diagram representing the quantification on the left of each graph. Number of junctions analysed in each condition is shown at the bottom of the graphs, below each scatter box plots. Junctions were obtained from biological replicates (N = 2). Statistical analysis was performed using Mann-Whitney U-test. Non-significant (ns) and significant p-values (<0.05) are placed inside graphs. Scale bar = 20 µM or 10 µM (zoom images).

The online version of this article includes the following source data and figure supplement(s) for figure 7:

**Source data 1.** Data from endothelial cell stimulation used in graphs in *Figure 7B–F*.

**Figure supplement 1.** Impact of different users on Junction Mapper quantification of endothelial junctions.

**Figure supplement 1—source data 1.** Impact of user-bias on Junction Mapper quantification of endothelial junctions.

tension in receptor modulation and integrity of multicellular tissues (*Liu et al., 2010*), and merit further experimentation.

Because of the fragmented and undulated nature of cardiomyocyte contacts (*Ehler, 2016*; *Vermij et al., 2017*), quantitative imaging tools specifically designed for intercalated discs have not been available or systematically used. At steady state, our data show that, in control cardiomyocytes, intercalated discs have clusters of cadherin receptors that are far apart. Hypertrophic stimulus *in cellulo* (neonatal rat cardiomyocytes) potently increases the levels of β-catenin at contacting interfaces, consistent with what was reported in hamster and human hypertrophic hearts (*Masuelli et al., 2003*). Connexin 43 is a major connexin isoform found in cardiomyocytes and its total protein and mRNA levels are augmented by hypertrophic signals *in cellulo* (i.e. phenylephrine) (*Salameh et al., 2008*; *Stanbouly et al., 2008*) or in human hearts with compensated left-ventricular hypertrophy by pressure-overloading (*Kostin et al., 2004*). The initial profiling analysis with Junction Mapper suggest that the formation of gap junctions is enhanced after hypertrophy stimulation *in cellulo*, confirming the broadening of intercalated disc area and higher number of GAP junctions in compensated hypertrophic hearts of human patients (*Kostin et al., 2004*). Although hypertrophic stimulus increases both β-catenin and connexin 43 levels at intercalated discs, these markers are regulated in distinct ways. The β-catenin cluster density is decreased leading to a more contiguous distribution, while connexin 43 is localised in clusters of higher density.

Clearly, further investigation is necessary to ascertain the profiling and biological significance of the phenotypes observed in different models. The quantification of very fragmented and zig-zagged junctions such as those in endothelia and cardiomyocytes is a challenge that Junction Mapper has begun to address, but improvements in future computation studies are welcomed. The complexity of these junctions also requires more user input to quality control the definition of the skeleton and corners. In addition, the ability of Junction Mapper to quantify junctions in a stratified epithelium, where the added complexity of multiple epithelial cell layers provides an additional challenge, remains to be tested.

We foresee the potential of Junction Mapper in distinct research areas, due to its innovative and in-depth approach to quantify cell–cell contacts. The detailed fine mapping of junction properties forms a basis to distinguish between disassembly mechanisms and infer cellular processes such as intracellular trafficking, receptor clustering or modulation of contraction at junctions. As multiple cellular processes contribute to junction stability, the fingerprinting of junction phenotypes after different stimuli is a powerful tool in pathway inference and guides rescue and translational experiments. Despite its limitations, Junction Mapper's broad dynamic range, repertoire of novel parameters and applicability to quantify junctions in various cell types will have a significant impact on studies in numerous model systems.

## Source data

Numerical data used to prepare graphs in each figure. Data for each graph are listed in separate sheets in the Excel files. Explanations can be found in the first sheet of each Excel file.

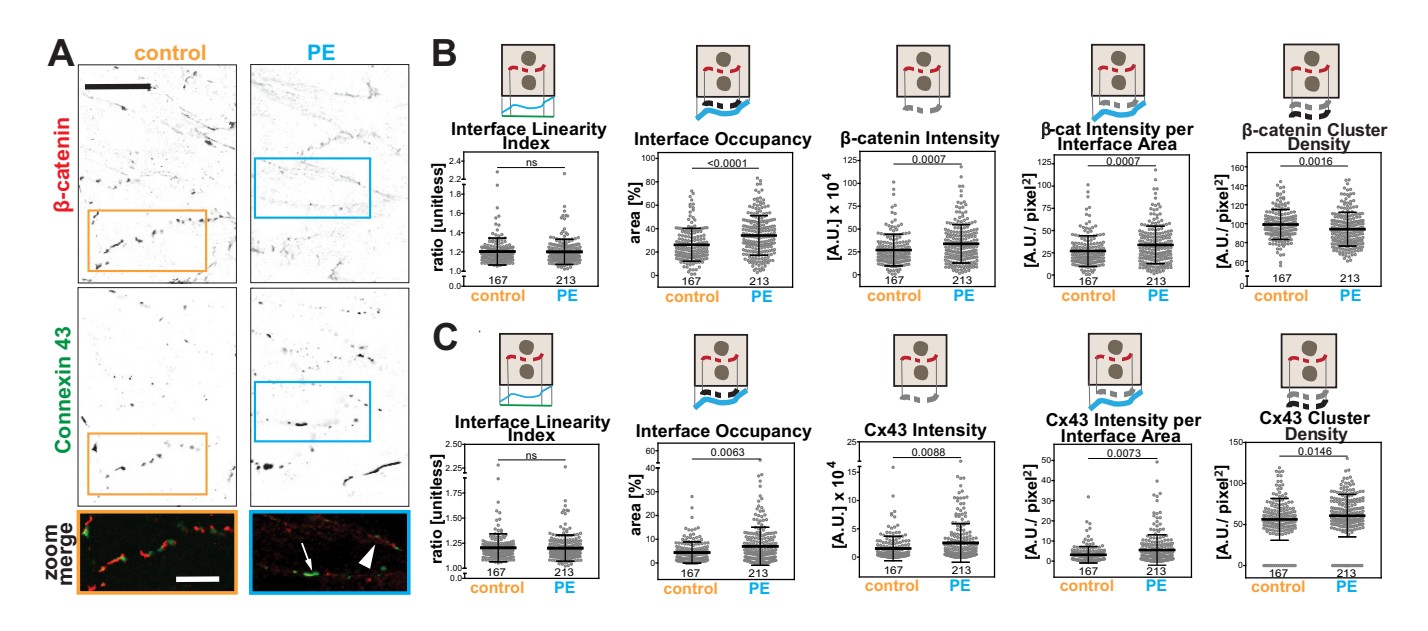

**Figure 8.** Hypertrophic stimulus of cardiomyocytes promotes distinct responses of Connexin 43 and β-catenin at intercalated discs. (**A**) Rat neonatal cardiomyocytes were stimulated with phenylephrine for 48 hr and stained with anti-β-catenin or anti-connexin 43 antibodies. Inverted images for each marker and a zoom of merged staining are shown. The arrowhead points to β-catenin staining and the arrow shows connexin 43 clusters. (**B-C**) Images were processed with Junction Mapper and quantification of selected parameters is shown here for β-catenin (**B**) or connexin 43 (**C**). Diagrams on top of each graph summarize how each parameter was measured. Values were obtained from one technical replicate. Number of junctions analysed in each condition is shown below each scatter box plot. Statistical analysis was performed using Mann-Whitney U-test. Non-significant (ns) and significant p-values (<0.05) are placed inside graphs. Scale bar = 10 µM.

The online version of this article includes the following source data for figure 8:

**Source data 1.** Data obtained from cardiomyocyte experiments used in *Figure 8* graphs.

# Materials and methods

## Key resources table

| Reagent Type | Designation | Reference | RRID | Catalogue number |
|---|---|---|---|---|
| Antibody | β-catenin (rabbit polyclonal) | Thermo-Fischer | RRID:AB_88035 | Cat# 71–2700 |
| Antibody | Connexin 43 | Millipore | RRID:AB_11210474 | Cat# MAB3067 |
| Antibody | anti-myc | Sigma | RRID:AB_439695 | Cat# B7554 |
| Antibody | E-cadherin (HECD1) | own hybridoma stock | | |
| Antibody | VE-cadherin (clone 75) | BioSciences | RRID:AB_2276073 | Cat# 610252 |
| Recombinant DNA reagent | pEGFP- Src Y527F | activated Src | | gift Prof M Frame |
| Recombinant DNA reagent | pRK5-myc H-Ras G12V | activated H-Ras | | (*Braga et al., 2000*) |
| Recombinant DNA reagent | pRK5myc- Rac1 Q61L | activated Rac1 | RRID:Addgene_12983 | (*Lamarche et al., 1996*) |
| Sequence based reagent | siRNA duplexes | CIP4, VAV2 and EEF1A | | (*Erasmus et al., 2016*) |
| Software | ImageJ | http://fiji.sc | RRID:SCR_002285 | |
| Software | GraphPad Prism | https://graphpad.com | RRID:SCR_002798 | |
| Software | Adobe Illustrator | http://www.adobe.com /products/illustrator.html | RRID:SCR_010279 | |

*Continued on next page*

*Continued*

| Reagent Type | Designation | Reference | RRID | Catalogue number |
|---|---|---|---|---|
| Software | Adobe Photoshop | https://www.adobe.com/products/photoshop.html | RRID:SCR_014199 | |
| Software | Rstudio | http://www.rstudio.com/ | RRID:SCR_000432 | |

## Cell culture and treatment

Primary keratinocytes were grown as described elsewhere (*Braga et al., 1997*). Cells were transfected with activated Src (pEGFP-Src$^{Y527F}$, 1 µg/ml for 8 hr) or constitutively active GTPases Ras (pRK5myc-Ras$^{G12V}$, 0.5 µg/ml for 8 hr) or Rac1 (pRK5myc-Rac1$^{Q61L}$, 0.5 µg/ml, overnight) using Jet-Prime (Polyplus). For RNAi experiments, normal keratinocytes were seeded in standard calcium medium as above (containing 1.8 mM $CaCl_2$), transferred to low calcium medium (0.1 mM $CaCl_2$ and foetal calf serum depleted from divalent cations) and grown until confluence (*Braga et al., 1997*). For replicate validation, cells were treated with non-targeting oligos, and then transfected with pRK5myc-Rac1$^{Q61L}$. RNAi transfection was performed with oligonucleotide duplexes (at 50 nM) to deplete EEF1A (72 hr), VAV2 or CIP4 (48 hr) using Interferin (Polyplus) or Metafectene transfection reagents (Biontex Laboratories GmbH) (*Erasmus et al., 2016*).

Pooled Human Umbilical Vein Endothelial Cells (HUVEC) from different donors (Lonza) were cultured in EBM-2 culture medium supplemented with EGM-2 bullet kit (Lonza). Human plasma derived thrombin (used at 0.2 U/ml, 10 min) and fibronectin were purchased from Sigma-Aldrich. For staining, cells were plated on coverslips coated with 3 µg/ml fibronectin.

Neonatal rat cardiomyocytes were freshly isolated from 1- to 3 day old Sprague-Dawley rats, using Neonatal Heart Dissociation Kit and the protocol provided by the company (www.miltenyibiotec.com/protocols, Miltenyi Biotec). Cells were plated on 13 mm laminin-coated glass cover slips and cultured in 199 Medium (M199, Sigma) supplemented with 10% neonatal calf serum (NCS, Biosera), 1% Vitamin B12 (Sigma), 1% L-Glutamine (Sigma), 0.5% antibiotics. On the day after plating, cardiomyocyte cultures were exposed to 10 µM phenylephrine for 48 hr to induce hypertrophy (*Miragoli et al., 2011*).

## Immunostaining and microscopy

Normal keratinocytes were fixed for 10 min with 3% paraformaldehyde and permeabilized with 0.1% Triton X-100 and blocked with 10% FCS for 10 min and stained as described (*Braga et al., 1997*). Cells were stained with anti-E-cadherin antibodies (mouse monoclonal HECD1) and anti-myc (mouse monoclonal 9E10, Sigma-Aldrich Cat# B7554, AB_439695 from RRID https://scicrunch.org/resources). DNA was labelled with DAPI (Sigma, 1:3000). Secondary antibodies were bought from Jackson ImmunoResearch. After treatment, control and hypertrophy-induced neonatal rat cardiomyocytes were fixed and stained with antibodies against β-catenin (rabbit polyclonal 1:50, Thermo-Fischer Cat# 71–2700, RRID:AB_88035) and connexin43 (mouse monoclonal 1:1000, Millipore Cat# MAB3067, RRID:AB_11210474).

A summary of the types of images used for different experiments, type of replicate and user-defined parameters is found in *Supplementary file 1*. Briefly, images were acquired on a Leica DM IRBE confocal (keratinocytes), Zeiss inverted LSM-780 (cardiomyocytes) or LSM-510 (keratinocytes) laser-scanning confocal (Carl Zeiss) using a 63x/1.4 Plan Apochromat objective or with an Olympus Provis BX51 microscope coupled to a SPOT RT monochrome camera using Simple PCI software (Hamamatsu, Japan; keratinocytes).

Endothelial cells (HUVEC) were fixed for 15 min in 4% paraformaldehyde, permeabilized with 0.4% Triton X-100 for 15 min and blocked in 2% BSA for 1 hr. Mouse monoclonal anti-VE-cadherin (clone 75, BD Biosciences Cat# 610252, RRID:AB_2276073) antibody and secondary fluorescence antibody (Molecular Probes) incubations were performed in 2% BSA for 1 hr. Coverslips were mounted in Mowiol/DABCO solution. Coverslips were imaged using an inverted wide-field microscope (NIKON Eclipse Ti) equipped with a 60 × 1.49 N.A. Apo TIRF (oil) objective and Luca-R EMCCD camera (Andor).

| Parameters | Endothelial cells | | cardiomyocytes | | | |
|---|---|---|---|---|---|---|
| | VE-cadherin | | β-catenin | | Connexin 43 | |
| | control | thrombin | control | PE | control | PE |
| Interface Linearity Index | ● | ● *ns | ● | ● *ns | ● | ● *ns |
| Interface Occupancy | ● | ● | ● | ● | ● | ● |
| Protein Intensity | ● | ● | ● | ● | ● | ● |
| Protein Intensity/ Interface Area | ● | ● | ● | ● | ● | ● |
| Cluster Density | ● | ● | ● | ● | ● | ● |

**Scheme 4.** Overview of distinct changes of cell–cell adhesion in thrombin-stimulated endothelial cells or hypertrophic cardiomyocytes. Endothelial cells (HUVEC) were treated with thrombin and cardiomyocytes stimulated with phenylephrine (PE) to induce permeability or hypertrophy, respectively. Parameters are normalised to controls (junctions from untreated cells) arbitrarily set as 100 (orange colour). Values are represented as circles of proportional sizes for junctions from stimulated cells (blue colour). Non-significant values are shown in light blue colour (ns).

## Software development

Junction Mapper is an open access, standalone and downloadable software developed in Java. The Junction Mapper code has been deposited in GitHub (license GNU GENERAL PUBLIC LICENSE; https://github.com/ImperialCollegeLondon/Junction_Mapper; copy archived at https://github.com/ elifesciences-publications/Junction_Mapper) and the software, instructions and its documentation can be downloaded from https://dataman.bioinformatics.ic.ac.uk/junction_mapper. The software uses algorithms that are mostly available open access, with some added innovation (i.e. fragmented length calculation). The novelty of Junction Mapper lays in the integration of distinct measurements, calculations of new parameters and consolidation of all parameters in a single system. In addition to junction measurements, we also developed a 'Nucleus Tool' to quantify the inter-nuclei distances of scattering epithelial cells – this tool will be described elsewhere.

The parameters and software to quantify junction phenotypes were built on the concept of our prior segmentation image analysis (*Erasmus et al., 2016*) based on intensity thresholding and whole image output obtained for the epithelial monolayer. Briefly, the labelling with a junctional marker (i.e. cadherin or a generic 'junction marker 1') is used to delineate the borders between cells and to form a mask to extract only the pixels found at contact sites for quantification. The mask can then be used to segment additional co-stained proteins that localise at junctions.

We address the bottlenecks in quantification of phenotypes that are not measurable by intensity only. Segmentation of images was performed to delineate where cell–cell contacts are (cell skeleton) and identify the corners between three or more contacting epithelial cells (see *Figure 2—figure supplement 1* and *2*). The cell skeleton is calculated in the following way. The original grey scale image of the 'junction marker 1' edges is interactively edited using blurring and sharpening filters to emboss the edges using the program GUI. The image is then binarized using an averaging filter (of the target pixel local neighbourhood) alongside a threshold and this binary image of cell edges is then super-imposed upon the original cell image stained for the junctional marker. Super-imposition of the skeleton and grey-scale image allows cell boundaries of each cell to be fine-tuned or re-

adjusted automatically using dilation, skeletonisation and maximum intensity within a neighbourhood algorithm. Edges can also be drawn on and removed from the image manually during this interactive process. The result is well-characterised edges that correspond to the real cell boundaries with a minimal amount of effort from the user. It has so far proved very hard to obtain reliable cell boundaries on a wide range of images and conditions without an interactive element. The program is designed to make this task as quick and efficient as possible. The image skeleton can then be dilated by the user (1 to 9 pixels, depending on image amplification and junction width) to set the area that fully contains the junctional marker staining. Corners between contacting cells are automatically identified and can be added or corrected manually. For junctions that are severely perturbed, providing a labelling for the whole cell facilitates the positioning of its corners and skeleton.

These spatial demarcations (skeleton, corners and area) are then used to calculate the primary parameters as described below (Figure S2). A minimal threshold is applied to avoid losing any signal at the cell–cell interface. The length and area of (i) contacting membrane between cells (Interface Contour (unit pixels) and Interface area (unit pixels$^2$), respectively) was calculated using the skeleton between two neighbouring corners on the cell outline. The length and area of the specific staining of a junctional marker (Junction Contour (unit pixels) and Junction Area (unit pixels$^2$), respectively) were measured between the outermost above-threshold pixels of the staining observed along the contacting interface.

To evaluate distinct perturbations of junctions, primary parameters calculated the sum of the length of individual staining fragments of the junctional marker along the interface between cells (Fragmented Junction Contour, unit pixels). The Euclidian distance between corners was used to calculate the length of an optimal, straight interface between two cells (Straight-line Interface Length, unit pixels). For estimation of the optimal junction length, the parameter Straight-line Junction Length is derived from the Euclidian distance between the outermost above-threshold pixels of the staining of a junctional marker, which may or may not coincide with the distance between the corners. Additional primary parameters quantify more specifically the junctional marker: 'junction marker 1' Area (number of above the threshold pixels within dilated edge area, in pixels$^2$) and 'junction marker 1' Intensity (within the dilated edge area, in arbitrary units or A.U.).

Secondary parameters were derived to assess different phenotypes of perturbed contacts by normalizing measurements per length or area of each junction or contacting interface (Figure S3). First, the software calculates how straight an interface or junction is using two parameters: (i) Interface Linearity Index (ratio between Interface Contour and Straight-line Interface Length, unitless) and (ii) Junctional Linearity index (ratio between Junction Contour and Straight-line Junction Length, unitless). Second, to estimate the proportion of the interface between cells that is covered by adhesion receptors, two parameters are calculated: (i) a length-based parameter, Coverage Index (ratio between Fragmented Junction Contour and Interface Contour, unit %) and (ii) an area-based parameter, Interface Occupancy (ratio between Junctional area and Interface area, unit %). The latter would be more appropriate to account for variable thickness of the junctional marker staining. Third, secondary parameters that address the specific distribution of junctional markers between neighbouring cells are: 'junction marker 1' intensity per interface area (ratio between 'junction marker 1' Intensity and Interface Area, unit A.U./pixel$^2$) and Cluster Density (ratio between 'junction marker 1' and Junction area, unit A.U./pixel$^2$).

## Software validation

Validation of the length and area measurements of the interface, junctions and fragmented junctions was performed on selected junctions. The impact of user-controlled settings (dilation and thresholding) was tested by increasing the dilation or thresholding values during the analyses and comparing the effects on primary parameters of selected junctions. The coverage index was validated manually using FIJI and the active Rac1 expressing data set, on the same junctions quantified by Junction Mapper. The manual calculation (*Lozano et al., 2008*) has been traditionally done using the Euclidian distance (straight-line length) rather than the more precise contours measured by Junction Mapper. To estimate the impact of different users subjectivity on the data obtained with Junction Mapper, a random subset of images from three different data sets (siCIP4, H-Ras, endothelial cells) was analysed independently by a separate user without knowledge of the settings used for analysis by user A. Each user independently set up the corners, skeleton corrections, dilation and thresholding.

For estimation of signal to noise ratio required for the skeleton recognition by Junction Mapper, random noise was added to the original (high quality) image using Fiji (Gaussian (normally) distributed with a mean of zero and standard deviation of 25; https://imagejdocu.tudor.lu/gui/process/noise) (*Ferreira and Rasband, 2012*). The quality of images was then estimated using the peak signal-to-noise ratio (PSNR) expressed in decibels (dB). The PSNR was calculated with the SNR Fiji plugin (*Sage and Unser, 2003*) by comparing original image to the images with added noise to the junction marker channel, since only this channel was used to create the skeleton. A heuristics approach on how to optimize analyses with Junction Mapper can be found in *Supplementary file 4*.

## Image analysis – quality control and exclusions

Junction Mapper quantifies images in a variety of formats and resolutions. A summary of the image dataset and user-controlled values used in each experiment can be found in *Supplementary file 1*. Images obtained from different experiments were subjected to quality control before quantification: junctions were excluded if they have blurry areas, artefacts or large gaps between cells. Regions were also excluded that contained (i) junctions of cells overlapping or on top of each other (different focal plane) (ii) multinucleated cells (iii) cells that were not fully surrounded by neighbours (i.e. at the border of the image or epithelial colony), (iv) junctions of cells overexpressing high levels of protein. As values are obtained per junction and a junction is shared by two cells, duplicated measurements of junctions are removed from the dataset.

For expression of different oncogenes, junctions were classified as those between (i) control cells (between two non-transfected cells), (ii) between two expressing cells (ee) or (iii) junctions shared by one expressing and a non-expressing cell (en). For the analyses of endothelial cells - junctions in blurry parts of the picture, with artefacts or large gaps between junctions and junctions at the border of a endothelial colony were excluded. For cardiomyocytes, the same criteria were applied as for endothelial cells and the parts of the image where the staining seemed very scattered and chaotic, with no clear trace of a junction were also excluded.

## Statistical analysis

Normality test was performed in each dataset using Kolmogorov-Smirnov normality test and Shapiro-Wilk test. Data from Src$^{Y527F}$, H-Ras$^{G12V}$ and Rac1$^{Q61L}$ experiments were analysed using ANOVA with the Games-Howell post-hoc test from the 'userfriendlyscience' in RStudio. Despite the data being non-parametric, the large number of junctions in each sample (>100 junctions) allows for the use of ANOVA with Games-Howell post-hoc test, which corrects for unequal sample sizes and variances between groups and for data with non-parametric distribution. Significance was set at $p < 0.05$. Data with a single treatment group and control group (this includes siRNA experiments in epithelial cells, cardiomyocytes and endothelial cells) were non-parametric and hence were analysed using the Mann-Whitney U-test in GraphPad PRISM. Pair-wise comparison was analysed with Wilcoxon matched paired test. The summary of the types of data used and statistical analyses can be found in the *Supplementary files 1* and *4*, respectively.

When biological replicates were analysed, it was first checked if the profile of the data was consistent across replicates, and then data was pooled. A total of 4080 junctions were quantified across different experiments (expression, siRNA or stimulus) and cell types using different batches of cells. An average of 227 junctions for each sample were analysed in parallel. The precise number of junctions quantified in each sample is written inside each graph. All numerical source data for each figure can be found in supplementary files online.

Graphs were obtained using GraphPad Prism. Images were processed using FIJI (*Schindelin et al., 2012*), Adobe Photoshop and Adobe Illustrator.

## Acknowledgements

The authors declare no competing interests. We acknowledge generous funding by Medical Research Council (MR/M026310/1, SB), Cancer Research UK (C1282/A11980, JE), BBSRC (BB/M022617/1, JE), National Institutes of Health Grant (1R01HL141855-01, JG), BHF (RG/17/13/33173, JG), Brunei Government Studentship (NM-N) and NHLI Foundation Studentship (MS). Netherlands Organization of Scientific Research (VIDI 016.156.327, SH). The Facility for Imaging by Light

Microscopy (FILM) at Imperial College London is part supported by funding from the Wellcome Trust (grant 104931/Z/14/Z) and BBSRC (grant BB/L015129/1).

## Additional information

### Funding

| Funder | Grant reference number | Author |
|---|---|---|
| Medical Research Council | MR/M026310/1 | Vania MM Braga |
| Biotechnology and Biological Sciences Research Council | BB/M022617/1 | Vania MM Braga |
| Cancer Research UK | C1282/A11980 | Vania MM Braga |
| Nederlandse Organisatie voor Wetenschappelijk Onderzoek | VIDI 016.156.327 | Stephan Huveneers |
| Prime Minister's Office, Brunei Darussalam | JPLL/A/4:A[2010]/J1(703) | Noor Mohd Naim |
| National Institutes of Health | 1R01HL141855-01 | Julia Gorelik |
| British Heart Foundation | RG/17/13/33173 | Julia Gorelik |

The funders had no role in study design, data collection and interpretation, or the decision to submit the work for publication.

### Author contributions

Helena Brezovjakova, Formal analysis, Validation, Investigation, Visualization, Writing - original draft, Performed the formal analyses, software validation and visualization of the data; Chris Tomlinson, Software, Supervision, Validation, Methodology, Writing - original draft, Designed the software and computational solutions, Produced the instructions video and website; Noor Mohd Naim, Data curation, Investigation, Methodology, Provided resources (image datasets); Pamela Swiatlowska, Formal analysis, Validation, Visualization, Methodology, Writing - original draft, provided resources (image datasets); Jennifer C Erasmus, Data curation, Supervision, Investigation, Visualization, Methodology, Writing - original draft, provided resources (image datasets) and software validation; Stephan Huveneers, Resources, Data curation, Validation, Writing - original draft, Writing - review and editing, Provided datasets; Julia Gorelik, Conceptualization, Supervision, Funding acquisition, Methodology, Project administration, Writing - review and editing, Provided datasets; Susann Bruche, Conceptualization, Software, Supervision, Investigation, Visualization, Methodology, Writing - review and editing, provided resources (image datasets) and software validation; Vania MM Braga, Conceptualization, Software, Funding acquisition, Writing - original draft, Project administration, Writing - review and editing

### Author ORCIDs

Helena Brezovjakova (iD) https://orcid.org/0000-0002-3554-6084
Chris Tomlinson (iD) https://orcid.org/0000-0002-8634-6111
Pamela Swiatlowska (iD) http://orcid.org/0000-0002-3705-7495
Stephan Huveneers (iD) https://orcid.org/0000-0002-1091-475X
Julia Gorelik (iD) http://orcid.org/0000-0003-1148-9158
Susann Bruche (iD) https://orcid.org/0000-0002-5814-7166
Vania MM Braga (iD) https://orcid.org/0000-0003-0546-7163

### Decision letter and Author response

Decision letter https://doi.org/10.7554/eLife.45413.sa1
Author response https://doi.org/10.7554/eLife.45413.sa2

## Additional files

### Supplementary files

- Source data 1. Data used to compare Coverage Index calculated by Junction Mapper and manually (*Supplementary file 2*).

- Source data 2. Junction Mapper data showing the impact of user-bias on the quantification of an epithelial dataset (*Supplementary file 3*).

- Source data 3. Data to demonstrate robustness of Junction Mapper quantification of biological replicates (*Scheme 1*).

- Supplementary file 1. Description of the different experiments used to validate Junction Mapper. Type of microscope, image characteristics, image resolution, user-controlled settings and replicate type are outlined.

- Supplementary file 2. Validation of the Coverage Index parameter. A-B Diagram and definition of the measurements used for quantification of the parameter Coverage Index. A, Junction Mapper measures the Coverage Index using the more precise contour length of E-cadherin fragments. B, Our previous work (*Lozano et al., 2008*) defined Coverage Index as the ratio between the straight lines (Euclidian distances) measuring E-cadherin staining over the interface length. C, Quantification of the same images is shown via the two methods, Junction Mapper or Manual. When manually quantified (i.e. straight line) the average values are smaller than those obtained with Junction Mapper. When the two methods were compared, there are no statistical differences between control groups or between active Rac1 groups. Furthermore, the significant difference between control and Rac1 is maintained in each methodology (manual or Junction Mapper). Junctions were quantified from one technical replicate; number of analysed junctions is written inside graphs below each sample. Ns, non-significant; ***p = 0.001. Scale bar = 10 μm.

- Supplementary file 3. Impact of user on Junction Mapper quantification of epithelial junctions. A-B A subset of representative images was processed by two users independently (first column; user A and user B), setting up the skeleton, corners, dilation, thresholding values. Middle column shows the overlay of the skeletons obtained by user A and user B; arrows point to misaligned regions. Last column shows the overlay of added corners to the skeleton. Blue arrowheads show corners that are not co-localized. A, Control samples (CIP4 siRNA experiment) were stained for E-cadherin (red) and F-actin (green). B, Epithelial cells expressing activated H-Ras (green, pRK5-myc-H-Ras$^{G12V}$) stained for E-cadherin (red). C, Pairwise comparison of individual junction values of primary parameters of CIP4 siRNA experiment obtained by user A (dilation 2, threshold 50) and user B (dilation 3, threshold 69). D, Pairwise comparison of individual junction values of primary parameters of active H-Ras expression images obtained by user A (dilation 2, threshold 54) and user B (dilation 2, threshold 55). E-F, Graphs of selected parameters obtained from the analyses by user A and user B. The overall result and profile comparing control and treated samples is similar between different users. E, RNAi experiment showing control non-targeting siRNA (NT) and CIP4 siRNA samples. F, Expression of activated H-Ras showing junctions from control non-expressing cells (c), between H-Ras expressing and non-expressing cells (en) and between two expressing cells (ee). Number of junctions analysed by each user is shown on the Y axis of first graphs (panels C,D) or below scatter plots inside the last graph on the right (panels E,F). All junctions are from one technical replicate. Statistical analyses were performed by Wilcoxson matched-pairs signed rank test (C-D) or One-Way Anova and Kruskal Wallis test (E-F). ns, non-significant

- Supplementary file 4. Heuristics approach to set up analysis with Junction Mapper and minimize user bias.

- Supplementary file 5. Description of statistical analyses and variances of the experimental data analysed by Junction Mapper.

- Transparent reporting form

### Data availability

The Junction Mapper code is licensed in github as GNU GENERAL PUBLIC LICENSE. The address is: https://github.com/ImperialCollegeLondon/Junction_Mapper (https://github.com/elifesciences-publications/Junction_Mapper). The software is downloadable as an executable jar file from https://

dataman.bioinformatics.ic.ac.uk/junction_mapper/. The image data used in this study has been previously published elsewhere (Erasmus et al., 2016; Huveneer et al., 2012) or are in preparation in separate mechanistic studies (Bruche et al., in preparation). Excel files of the output of parameters and calculations has been provided as source data files online.

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

## Appendix 1

# Junction Mapper: user notes and algorithm definitions

## Introduction

Junction Mapper is a semi-automated software application for analysing data from images of cells in close proximity to each other in monolayers. The software can be obtained as described below:

- Downloadable as an executable jar format: https://dataman.bioinformatics.ic.ac.uk/junction_mapper/
- Github address: https://github.com/ImperialCollegeLondon/Junction_Mapper (copy archived at https://github.com/elifesciences-publications/Junction_Mapper)
- License: GNU GENERAL PUBLIC LICENSE

The focus of Junction Mapper is to measure the morphology of cell boundaries, define single junctions and quantify the length, area and intensity of the staining of different proteins localised at cell-cell contacts. The output are various unique parameters that assess the contacting interface between cells and up to two junctional markers. Here we describe the operational mode of the software and how the different steps and parameters are calculated computationally.

## Image Loading

Junction mapper is suitable for analysis of cell images that have the following properties:

- The system will load single images;
- several images can be opened in the program at the same time in individual analysis tabs
- the tiff format is the most optimal highest image quality
- The images can have up to three channels that correspond to the following features;
  - Junction marker 1 – used to set up the cell boundaries and measurements
  - Junction marker 2 – additional marker that can be measured.
  - Cell nuclei (optional)

Upon loading an image, the user is asked to define the channels in the tiff image where the three expected image analysis components are located (Junction 1 Channel, Junction 2 Channel and Nucleus). These can be defined in Junction Mapper for images stained with any combination of fluorescent conjugates. Users may add notes to their analysis in the text box below the channel dropdown menus. These will appear with output materials.

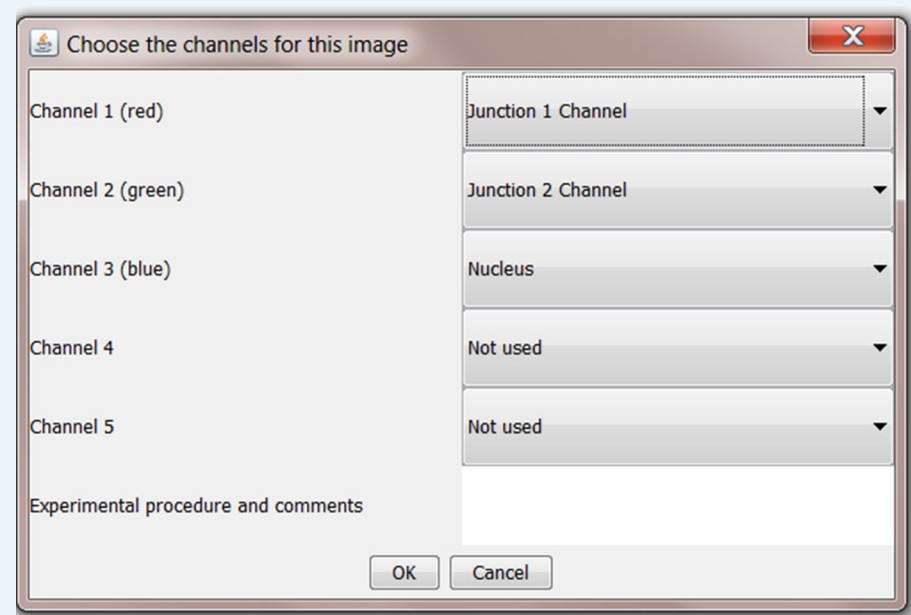

**Appendix 1—figure 1.** Screenshot of allocation of image channels to selected analyses components for processing.

## Data Output

When Junction Mapper is started by the user, an output directory for the analysis data is automatically created as a subdirectory of the directory location where Junction Mapper was started from. This directory is named in the following format <dd > _<mm > _<yyyy > _hhmm, so if Junction Mapper was started on 12th April 2019 at 09:18 the output directory created would be called; 2019_04_12_0918. Data is saved by Junction Mapper when the 'Save Edge Map' or 'Save as Spreadsheet' controls are used.

## General Operation

Once the cell image is loaded, the user should see the corresponding image (example below). There are six tabs:

- four of which contain the original composite image and the three split channels, respectively, and
- two tool tabs (Junction 1 Tool and Nucleus Tool). These two tool tabs operate independently of one another.

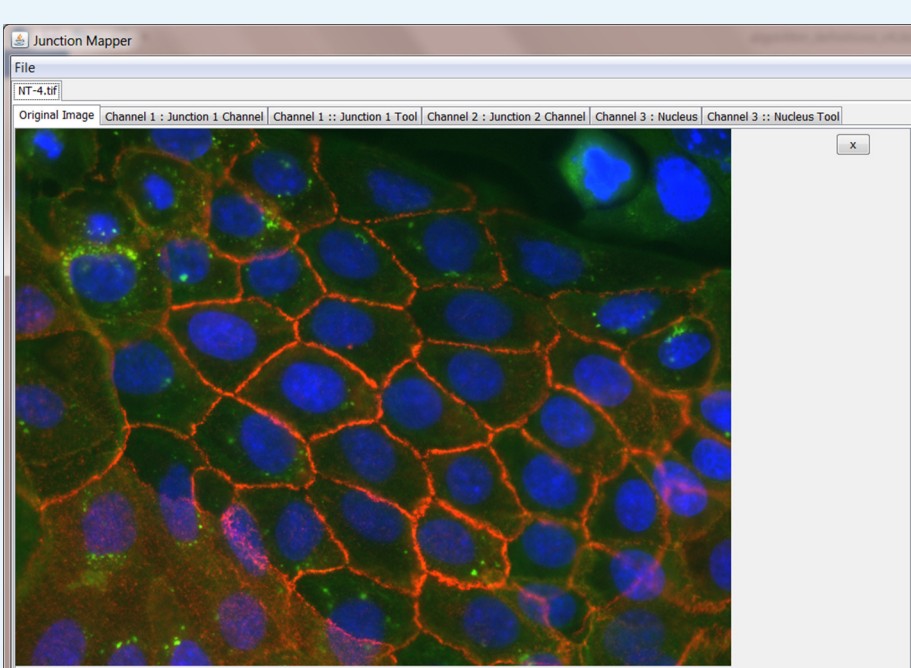

**Appendix 1—figure 2.** Screenshot showing loaded merged image and corresponding tabs for analyses.

## Junction 1 Tool

This tool is used to first generate an edge map reflecting the cell junction outline and then to measure distinct parameters at the cell boundary using the Junction one and the Junction 2 channel of the original image. To use the Junction one tool, select the appropriate tab in the image. In order to obtain the edge map and use it to calculate the junction parameters, a number of steps are performed via the Junction one tool as outlined below:

- Step 1: Edge Detection
- Step 2: Produce Binary Edge Map
- Step 3: Finesse the edge map
- Step 4: Select cells to be analysed
- Step 5: Select Individual Cell to Analyse
- Step 6: Define corners of cell
- Step 7: Measure different parameters at cell-cell contacts

## Step 1: Edge detection

In order to get meaningful results from Junction Mapper, it is very important to construct an accurate representation of the cell boundary. Cell images can be taken with varied magnifications and image quality. Junction Mapper is sufficiently flexible to perform well on many different image types. The first step is to emboss the edges of the cells by using the tools on the panel. There are three filters on the right side panel:

- **Gaussian** - blurs cell edges
- **Median** - makes edges more uniform and removes spot noise
- **Sharpen** - makes edges stand out more

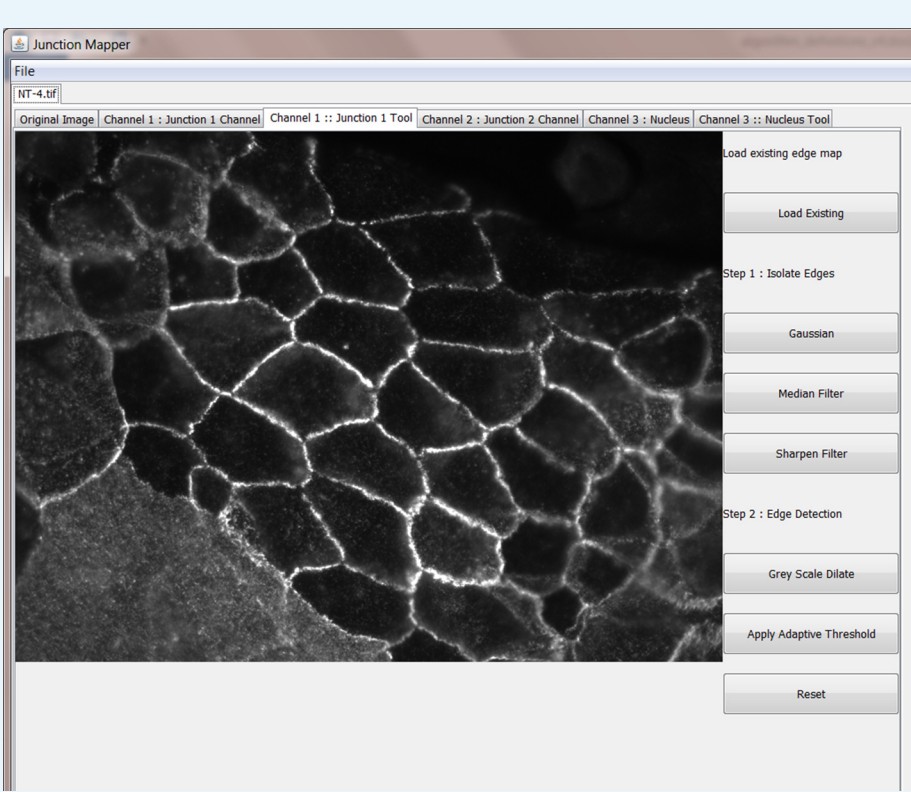

**Appendix 1—figure 3.** Screenshot of the edge detection function main menu.

## Step 2: Produce Binary Edge Map

The Grey Scale Dilate function can be used to fill any holes in the cell edge detected. The number of repetitions of these filters and the order and combinations that they are applied in is user-controlled and should be guided by the resultant image itself, which is displayed after every operation. Once the edges are well defined, the user should select the Apply Adaptive Threshold, which will binarize the image. The C value and filter size parameters to this operation are chosen from a dialog box that appears when this control is selected. The C value chosen depends on the quality and contrast of the image (higher contrast, higher C value), whilst the size of the filter selected should be slightly bigger than half of the average edge width in pixels. The result of the Apply Adaptive Threshold function is a binary image that is used as the basis to build the edge map. Existing edge maps saved previously can also be loaded into the system at this stage.

| Control name | Algorithm used |
|---|---|
| Gaussian | Image is convolved with a 5 × 5 Gaussian kernel with values: {{1,4,7,4,1},{4,16,26,16,4},{7,26,41,26,7},{4,16,26,16,4},{1,4,7,4,1}}; |
| Median Filter | Median Filter applied to image with a 5 × 5 kernel |
| Sharpen Filter | Image is convolved with a 3 × 3 sharpening filter, kernel has values: {{−1,−1,−1},{−1,12,−1},{−1,−1,−1}} |
| Grey Scale Dilate | Performs a grey scale dilation operation on the edge image in a 3 × 3 neighbourhood by replacing the target pixel with the largest grey scale value in the local neighbourhood |

*continued*

| Control name | Algorithm used |
|---|---|
| Adaptive Threshold | Binarizes the grey scale edge image by using an adaptive thresholding technique. The user chooses a C value (range [0:50]) and a filter size from the set: {'3 × 3', '5 × 5', '7 × 7', '9 × 9', '11 × 11', '21 × 21', '35 × 35', '51 × 51','75 × 75','99 × 99'} The size of the window should be large enough to contain pixels of the structure being detected and background pixels. A window of the chosen filter size then calculates the average pixel intensity in the window for every pixel in the image and adds the chosen C value to it. If the target pixel original grey scale value is equal to or exceeds this value (average window intensity value + C value), then the target pixel value in the resultant binary image is set to one otherwise it is set to 0. |
| Reset | Returns to step 1 'Isolate edges' |

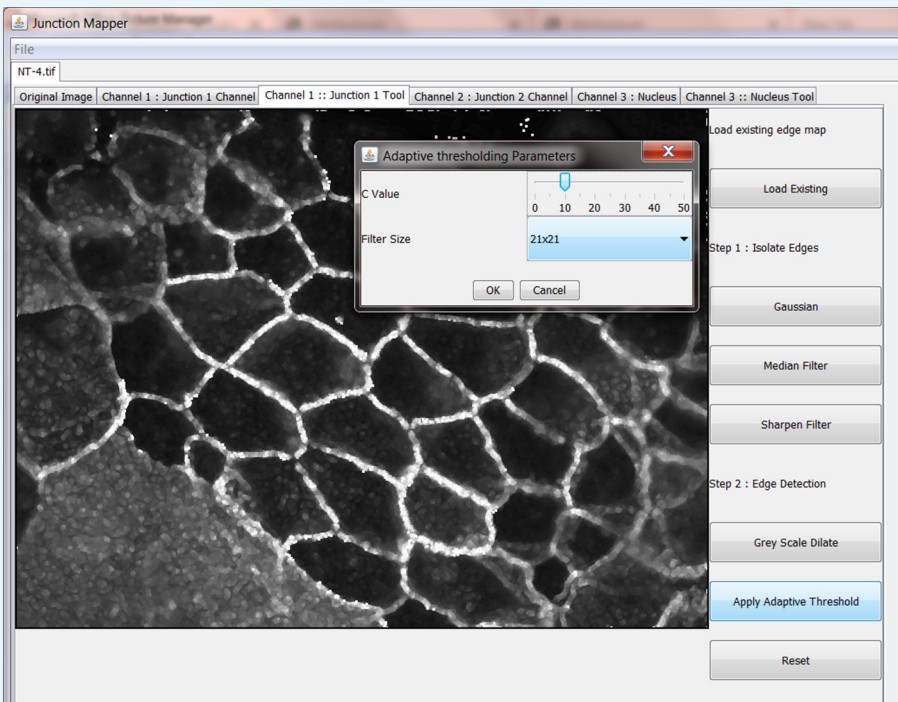

**Appendix 1—figure 4.** Screen shot of how to apply the Adaptive Threshold in images.

## Step 3: Finesse the Edge Map

At the beginning of this stage we have a binary image loosely corresponding to edges in the image that needs to be turned into an accurate edge map for further processing. The first step is usually to dilate the image two or three times (to join the detected edges) and then skeletonise (to create a single pixel wide edge). Trailing Edges and Remove Small Objects can be used to removed imperfections and misalignments in the binary image. The user should then check the accuracy of the edge map by using the Edge Map Toggle function to see how closely the map follows the edges in the original image. This function rotates the background on which the skeleton is drawn between blank, Junction one channel and Junction two channel allowing the user to assess the accuracy of the skeleton produced. Minor adjustments can be made by using the Local Maxima for Edges function and then Dilating (to join edges) and Skeletonise the result to get a single pixel wide edge. The Local Maxima for Edges can be used to move the edge to the brightest grey scale values in the vicinity of the current edge, based on the grey scale values in the original grey scale Junction one image. The local neighbourhood of the original Edge image is inspected around each skeleton pixel location and a '1' binary value assigned to the maxima in this window. This

operation can be performed on decreasing neighbourhood sizes in conjunction with the dilate and skeletonize functions to build a more accurate edge map. Edges can also be manually added or removed by the user by selecting the Add or Remove check buttons and clicking on the image itself. Click left to add a point and then click again to add a line between the points. This can be repeated to rapidly create an edge. Click with the right-hand button to stop the edges being added. Some of the functions will only work when the binary image is displayed (dilate, local maxima). When the edge map corresponds to the edges of the cells in the image (cells to be measured) click the **Finish** button. Completed edge maps can be saved at this stage for future use (Save Edge Map).

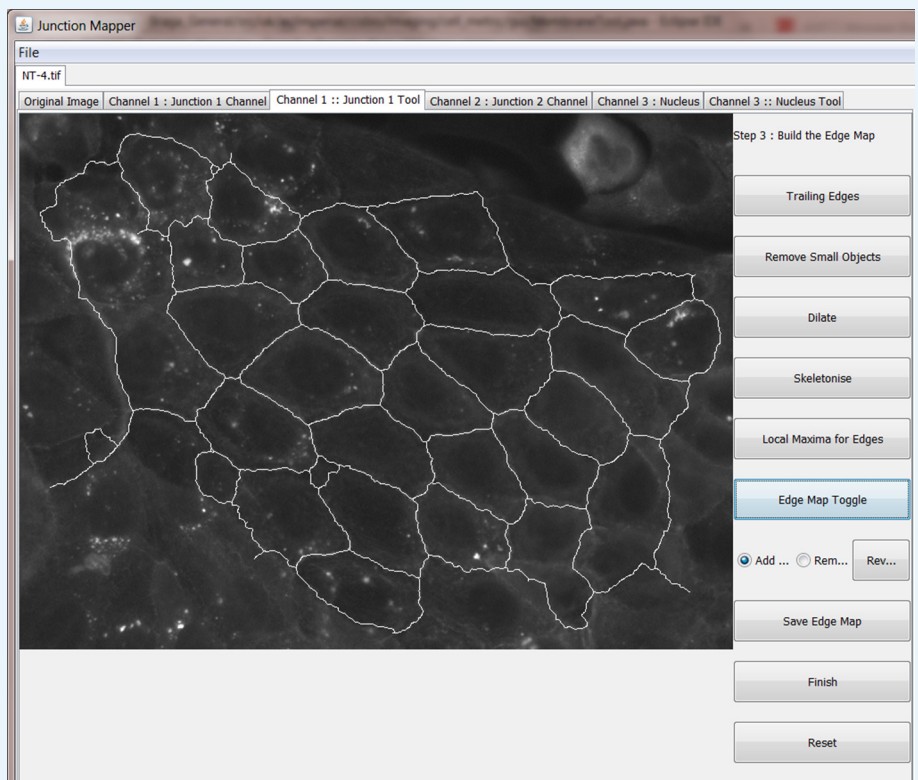

**Appendix 1—figure 5.** Screen shot of how to create the edge map or skeleton setting up cell boundaries.

| Control name | Algorithm used |
|---|---|
| Trailing Edges | Removes trailing edges from the image. This uses a bespoke algorithm written by the author and uses the binary edge map as input. First, the image is scanned to locate edge pixels that have only one neighbour and adds these to a list. Then, for each point in the list; 1. Remove the pixel from the target image 2. Check the neighbourhood for connected pixels, if there is only one connected neighbour add it to the list. This has the effect of removing trailing edge fragments from the edge map image. |
| Remove Small Objects | Removes pixel connected objects from the binary image that are smaller than a user selected threshold. User can choose from the set: {'10','20','30','40','50','100','150','200','250','300','350', '400','450','500'} Objects smaller than the selected threshold will be removed from the image. |
| Dilate | Standard binary image dilation algorithm |

*continued*

| Control name | Algorithm used |
|---|---|
| Skeletonise | Skeletonises image by using an adaption of the algorithm proposed in **Arcelli et al. (1975)**.<br>The masks proposed in the paper are applied in (simulated) parallel fashion to image pixels until the resultant image reaches idempotence. |
| Local Maxima for Edges | Alongside dilation and erosion, this operation is used to align the single pixel-wide edges to the junction on the original grey scale image (junction marker 1) more closely. It uses the original grey scale image and the single pixel-width edge map as inputs.<br>The user is asked to select the neighbourhood size from the set: {'3','5','7','9','11','15','21','25'}<br>For each pixel in the binary edge map that is set to a 1, the neighbourhood around that pixel is inspected in the original grey scale edge junction marker one image. The location of pixel with maximum grey scale value in the neighbourhood of inspected<br>pixel in junction marker one image is set to a '1' in the resultant binary image. It can be performed on gradually reducing neighbourhood sizes. |
| Edge Map Toggle | Rotates the background image that the binary edge map image is projected onto. Either shows just a binary image, the Junction one channel or the Junction two channel.<br>This control will rotate through the options in turn. |
| Add Straight Line | A tool to draw straight lines on the edge map image using the mouse.<br>Select by clicking the appropriate radio button and then click left mouse button on the image at the line start point and click again with the left button at the line end point. The line will be displayed on the image. Multiple straight lines can be added by multiple mouse clicks that uses the last point as the line start point. Edges can be traced quickly using this feature. Turn off the tool by clicking the right mouse button and move to a different location. |
| Remove Edge | A tool to erase parts of the edge map using the mouse. Select the appropriate radio button on the interface and move the mouse over the edge fragments that are to be removed. |
| Reset | Returns to step 1 'Isolate edges' |

## Step 4: Select cells to be analysed

The cells to be analysed are selected in this stage by clicking inside the cell body (the edge map for each selected cells must have closed edges for this to work). Each cell is labelled with a random colour and a unique number (in this image). At this stage, a model is made of the cell and its contour by the software. When all the cells to be analysed have been identified the Finish button should be pressed.

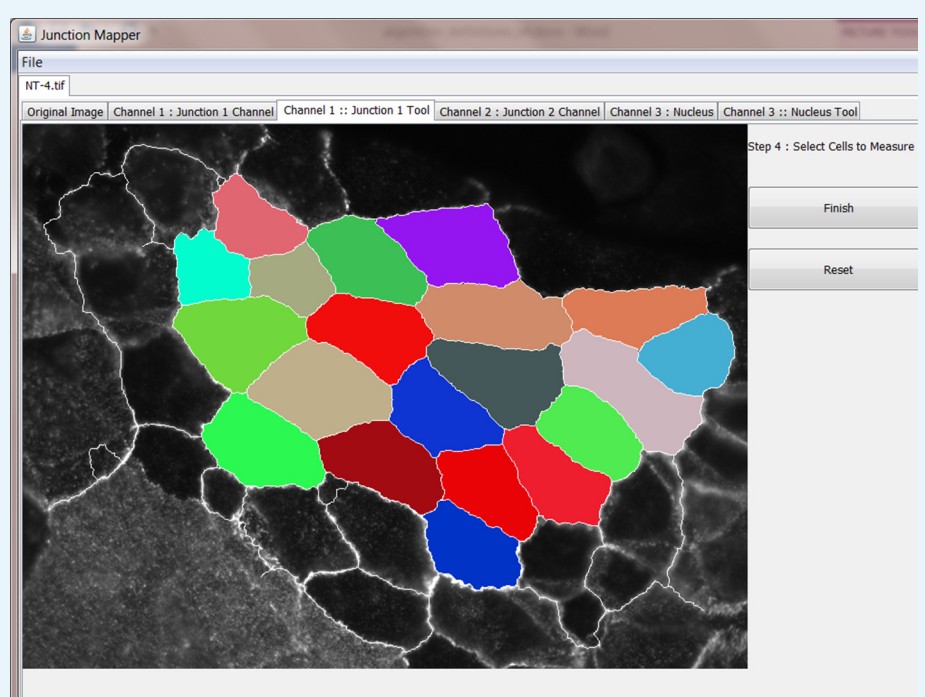

**Appendix 1—figure 6.** Screenshot demonstrating manual selection of all cells to be analysed by Junction Mapper.

| Step | Algorithm used |
| --- | --- |
| Region Growing | Users select the cells to be measured by clicking on them with the mouse. Using the mouse click point as a seed, a region growing algorithm uses the edge map as its boundaries and identifies the pixels contained within the cell body. |
| Reset | Returns to step 1 'Isolate edges' |

## Step 5: Select Individual Cell to Analyse

Each cell is automatically numbered. The cells are presented to the user and the user can choose which cell to analyse by clicking anywhere inside its boundary.

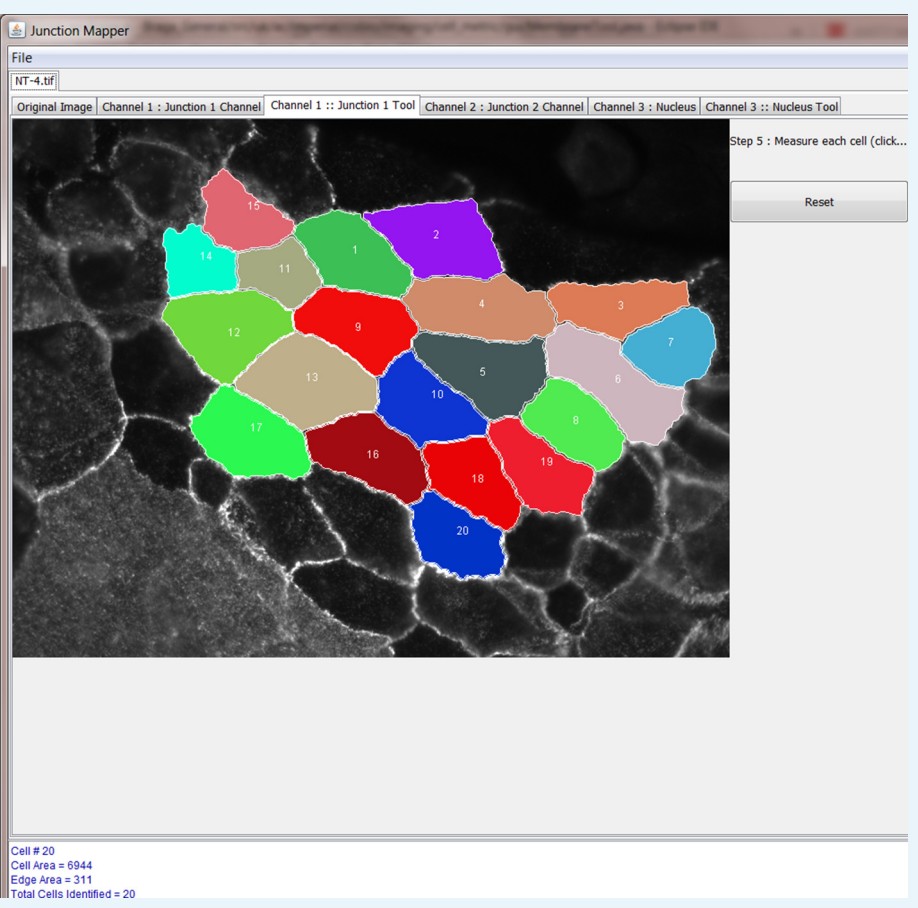

**Appendix 1—figure 7.** Screenshot of the automatic labelling of each cell to be analysed.

| Step | Algorithm used |
| --- | --- |
| Select Cell | User selects a cell to measure by clicking in its interior |
| Reset | Returns to step 1 'Isolate edges' |

## Step 6: Define Corners of Cell

At this step, the corners of a cell are defined for analysis. Pulling the top slider (Epsilon Value for Corner Detection) to the right will cause corners to appear in suggested places on the cell boundary as numbered yellow squares. The user can keep none, all or some of these corners. Existing corners can be removed by right clicking in the yellow square, new corners can be added by left clicking on the cell boundary position. The area to measure the junction marker 1 (used to make the edge map) is altered by using the Number of dilations slider: it sets by how many pixels the cell edge will be dilated on either side to form the area to be analysed. When this slider is changed, the area of the interface around the edge is projected onto the image in green with the edge represented as red contour line through the centre of the green area. Global thresholds for the measurements to be done in junction marker one image and junction marker two image can be set by the bottom two sliders. Only pixels with an intensity above the thresholds set for these channels are used in the calculation of some parameters (see definitions below). The Switch Background button changes the background image onto which the binary edge map and corner image is projected for the cell. The Measure button quantifies the parameters for the selected cell.

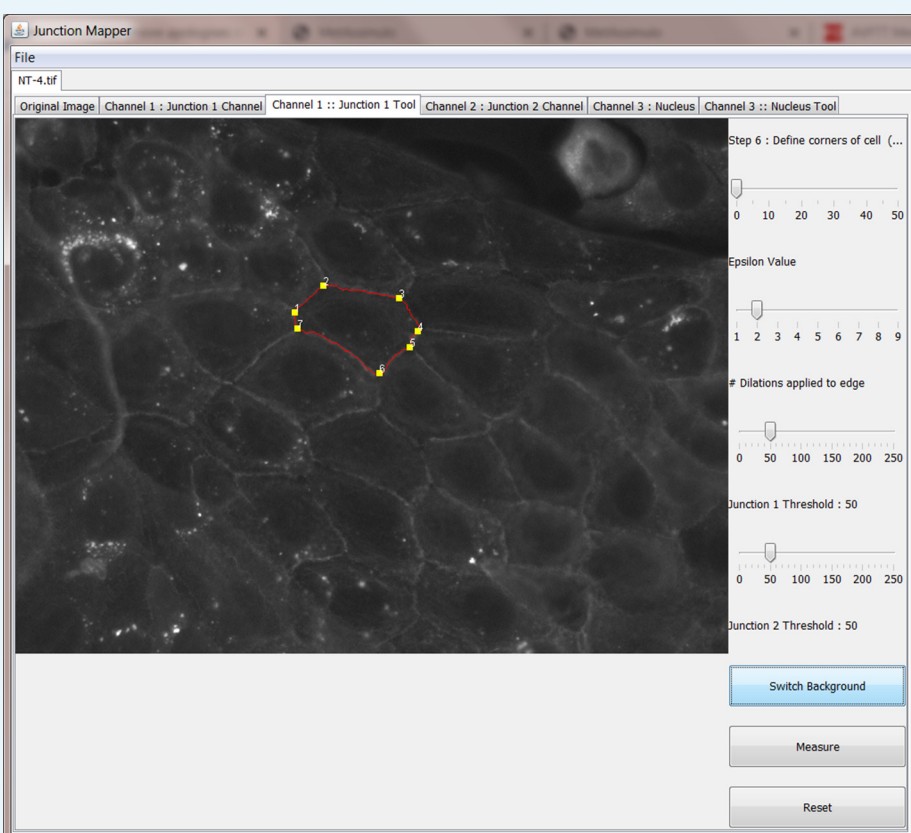

**Appendix 1—figure 8.** Screenshot of the menu to define cell corners automatically and manually.

| Control name | Algorithm used |
| --- | --- |
| Corner Detection | The algorithm used is based on the Douglas Peucker method defined in *Poiker and Douglas (2011)*.<br>The user sets the epsilon value required using a slider and the corners appear in the image as defined by the algorithm. Raising the epsilon value decreases the number of corners that the algorithm adds to the image. Corners can also be manually added (left mouse click on cell edge) or removed (right mouse click on corner) from the image. |
| Number of Dilations applied to Edge | This refers to the number of standard binary image dilations applied to the single pixel cell edge and creates the area will be measured. The area measured for the 'junction marker 1' and 'junction marker 2' by Junction Mapper can be altered using a slider control, the default value being 2. |
| Junction marker 1 Threshold | A global threshold applied to the 'junction marker 1' image over which pixels will be considered for inclusion in the resultant parameter calculations. |
| Measurement Threshold | A global threshold applied to the Junction two channel over which pixels will be considered for inclusion in the resultant parameter calculations. |
| Switch Background | There are three options: (i) just an edge map projected on to the 'junction marker 1' image,<br>(ii) the edge map projected onto the Junction two channel or<br>(iii) the edge map plus the 'junction marker 1' measurement area (the dilated edge)<br>projected onto the 'junction marker 1' channel. Using this control will rotate through the options in turn. |
| Measure | This button will perform the cell boundary measurements detailed elsewhere. Note that this operation does not save the measurements. |

*continued*

| Control name | Algorithm used |
| --- | --- |
| Return | Returns to step 5 'Measure each cell' |

## Step 7: Measure different parameters at cell-cell contacts

A visual representation of the measured area can be displayed edge by edge by using the Show Edge button. The resultant measurements and parameters calculated are saved by pressing the Save as Spreadsheet button. The data is saved as *.xls spreadsheet file in the output directory described earlier in this document. Along with the spreadsheet, several other files are saved in the output directory including image files and pdf documents containing the images produced. To return to the map of selected cells (shown in step 6), press the Back to Cell State button. Another cell can then be selected for processing in the same way. This step should be repeated until all cells have been measured.

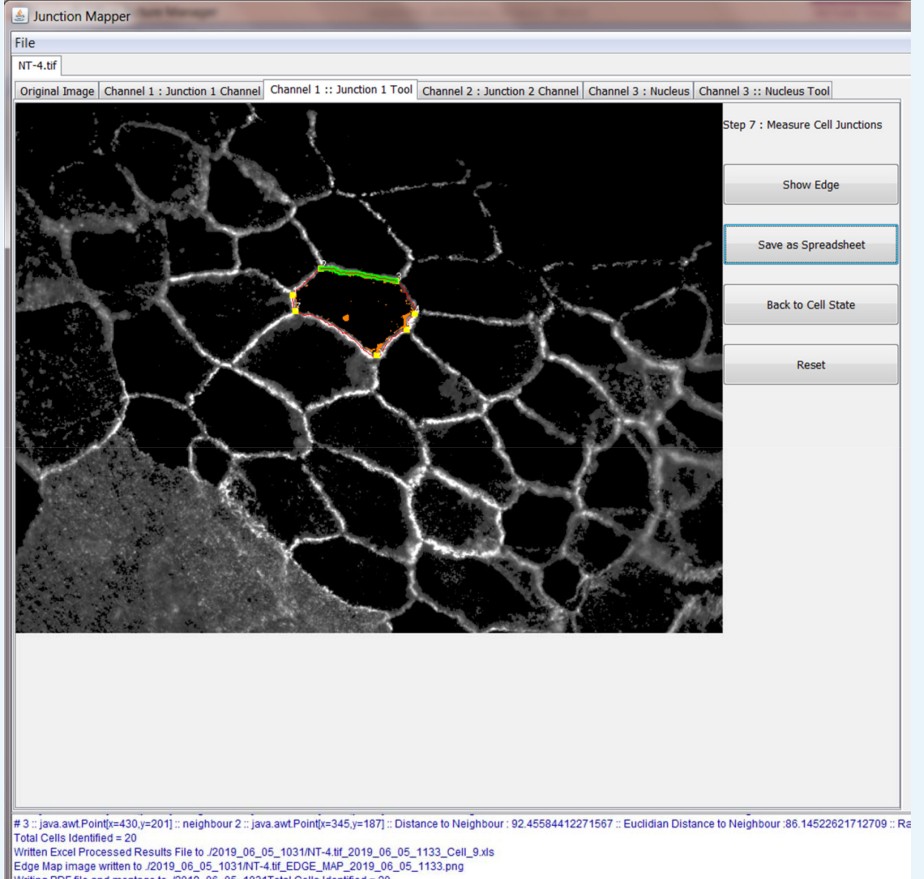

**Appendix 1—figure 9.** Screenshot of the menu to perform the measurements.

| Control name | Algorithm used |
| --- | --- |
| Show edge | Shows the edges measured individually (i.e. each contacting interface) projected onto the channel one image. Edge is displayed in red, whilst the 'junction marker 1' area measured is displayed in green. Pixels in the 'junction marker 1' channel that exceed the chosen threshold but are outside of the 'junction marker 1' area are displayed in orange and they are not computed in the parameters. |

*continued*

| Control name | Algorithm used |
|---|---|
| Save spreadsheet | Saves the junction parameters calculations in a spreadsheet for the selected cell along with corresponding reference images and a pdf document. The thresholded pixels outside the dilated area are also measured and recorded as Internal junction marker one area or Internal junction marker two area above the table of junction parameters in the spreadsheet. |
| Back to Cell State | Returns to previous step so that another cell can be selected for measurement |
| Return | Returns to step 5 'Measure each cell' |

The image below shows the spreadsheet produced by the process described here. Both the Junction marker one image and Junction marker two images are analysed simultaneously. The parameters calculated for each cell-cell contact are defined in the next section.

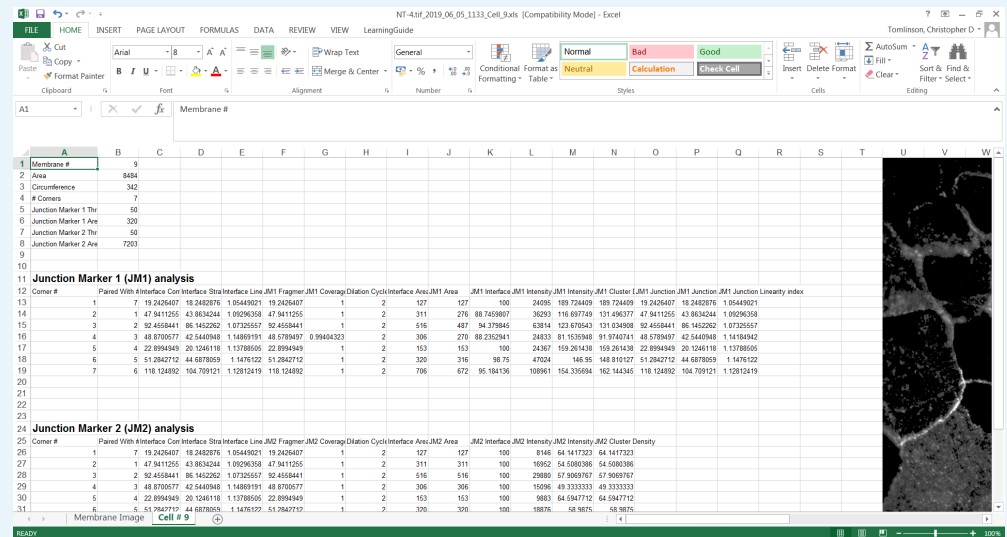

**Appendix 1—figure 10.** Representative image of the output of Junction Mapper as an excel file.

## Parameter Definitions and Formulae

This section defines the mathematical formulae to calculate the primary parameters for each edge identified in a cell. Secondary parameters are calculated by normalizing values with respect to the length or area of an interface or cell-cell contacts as outlined in the Suppl *Figure 4*. These measurements are output to the Excel spreadsheet as defined above.

| # | Name | Units | Description | Mathematical formula |
|---|---|---|---|---|
| 1 | Interface Contour | [pixels] | Distance between two corners of the defined cell edge | $L_E = \sum\limits_{i=1}^{n} \sqrt{(x_i - x_{i+1})^2 + (y_i - y_{i+1})^2}$ Where $L_E$ is the interface contour length, $p(x_i, y_i)$ is an edge pixel from an ordered list of edge pixels with cornerpixel1$(x_1, y_1)$ and cornerpixel2$(x_{n+1}, y_{n+1})$ |

*continued*

| # | Name | Units | Description | Mathematical formula |
|---|------|-------|-------------|----------------------|
| 2 | Straight-line Interface Length | [pixels] | Straight line distance between two corner points | $L_S = \sqrt{(x_1 - x_2)^2 + (y_1 - y_2)^2}$ <br><br> $L_s$ is the straight-line interface length where $p(x_1,y_1)$ is the first corner pixel on the cell edge $p(x_2,y_2)$ is the last corner pixel on the cell edge |
| 3 | Fragmented Junction Contour | [pixels] | Sum of stained fragments along the single pixel edge | $L_F = \sum_{j=1}^{n} \sqrt{(x_j - x_{j+1})^2 + (y_j - y_{j+1})^2}$ <br><br> Where $L_F$ is the fragmented junction contour length as the sum of the contour length of each junction fragment, $p(x_j,y_j)$ is an edge pixel from an ordered list of edge pixels with $p(x_1,y_1)$ being the first pixel on the cell edge fulfilling $I(p(x_j,y_j)) > \Theta_I$ AND $p(x_{n+1},y_{n+1})$ being the last pixel on the cell edge fulfilling $I(p(x_j,y_j)) > \Theta_I$, where $\Theta_I$ is the a priori defined intensity threshold, and $I(p(x_j,y_j))$ being the intensity of a given pixel Fragments must be at least two pixels |
| 4 | Dilation Cycles | [unitless] | Number of cycles used to dilate the defined edge | Number of times the binary image dilate algorithm is used to expand the defined edge. Essentially one dilation cycle changes a line of pixel width one to a line of pixel width 3. Two dilation cycles make the line five pixels-wide, etc. |
| 5 | Interface Area | [pixels$^2$] | Area in pixels of the dilated edge area between two corners | $A_E = \sum_{i=1}^{o} (A_p)_i$ <br><br> Where $A_E$ is the interface area as the sum of pixel area $A_p$ of all pixels in the dilated edge area and where $A_p$ is the pixel area with $A_p = 1px*1px = 1px^2$ and $o$ is the number of pixels in the interface area |
| 6 | Junction marker 1 Area | [pixels$^2$] | Area covered by junction marker staining within the interface area | $A_{FT} = \sum_{k=1}^{m} (A_p)_k$ <br><br> Where $A_{FT}$ is the junction marker area as the sum of pixel area $A_P$ of all pixels fulfilling the conditions $p(x_k,y_k) \in A_E$, AND $I(p(x_k,y_k)) > \Theta_I$ and where $A_P$ is the pixel area with $A_P = 1px*1px = 1px^2$ and $m$ is the number of pixels in the fragmented interface area $A_{FT}$ |

*continued*

| # | Name | Units | Description | Mathematical formula |
|---|------|-------|-------------|----------------------|
| 7 | Junction marker 1 Intensity | [A.U.] | Sum of cadherin (junctional protein) Intensity within the interface area. | $I_{FT} = \sum_{k=1}^{m} I_k$ <br> Where, $I_{FT}$ is the junction marker intensity as the sum of intensities $I_k$ of all pixels fulfilling the conditions 1) $p(x_k,y_k) \in A_E$, AND 2) $I(p(x_k,y_k)) > \Theta_I$ with m being the number of pixels in the fragmented interface area $A_{FT}$ |
| 8 | Junction Contour | [pixels] | Sum of pixel distances between the first and last junction marker pixels along the interface contour | $L_j = \sum_{j=1}^{n} \sqrt{(x_j - x_{j+1})^2 + (y_j - y_{j+1})^2}$ <br> Where $L_J$ is the junction contour length and $p(x_j,y_j)$ is an edge pixel from an ordered list of edge pixels with $p(x_1,y_1)$ being the first pixel on the cell edge fulfilling $I(p(x_k,y_k)) > \Theta_I$ AND $p(x_{n+1},y_{n+1})$ being the last pixel on the cell edge fulfilling $I(p(x_k,y_k)) > \Theta_I$ |
| 9 | Straight-line Junction Length | [pixels] | Euclidian distance from first to the last pixel of junction marker one on the interface contour | $L_{JS} = \sqrt{(x_1 - x_2)^2 + (y_1 - y_2)^2}$ <br> $L_{JS}$ is the straight-line junction length where $p(x_1,y_1)$ is the first pixel on the edge fulfilling $I(p(x_1,y_1)) > \Theta_I$ and $p(x_2,y_2)$ is the last pixel on the edge fulfilling $I(p(x_2,y_2)) > \Theta_I$ $\Theta_I$ is the a priori defined intensity threshold, and $I(p(x_j,y_j))$ being the intensity of a given pixel |

## Nucleus Tool

The nucleus tool allows users to count nucleus in an image and to measure the distance between them, which can be used to infer distance between neighbouring cells. This tool is useful as an indirect measurement of cell scattering. Three steps are performed via the **Nucleus Tool** tab and outlined below:

- Step 1: Adaptive Thresholding
- Step 2: Tidy Nucleus Image
- Step 3: Measure Distance between Neighbouring Cells

### Step 1: Adaptive Thresholding
The first step is to binarize the image via Apply Adaptive Threshold.

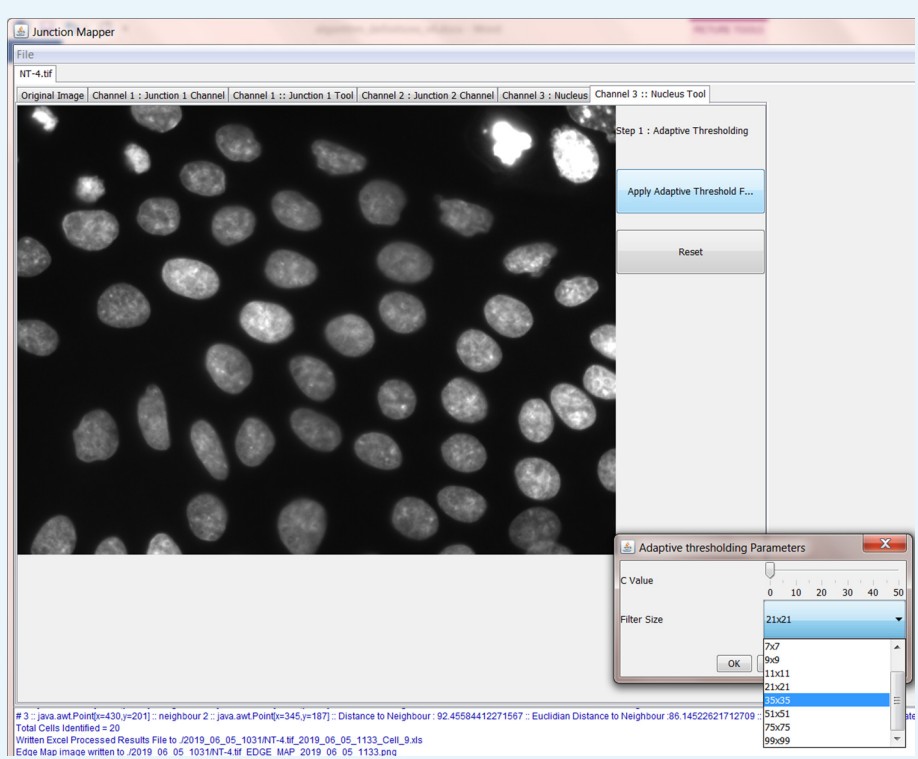

**Appendix 1—figure 11.** Table describing the parameters measured by Junction Mapper: diagram, units, definition and mathematical calculations.

| Control name | Algorithm used |
|---|---|
| Adaptive Threshold | Binarizes the grey scale nuclei image by using an adaptive thresholding technique. The user chooses a C value (range [0:50]) and a filter size from the set: {'3 × 3', '5 × 5', '7 × 7', '9 × 9', '11 × 11', '21 × 21', '35 × 35', '51 × 51','75 × 75','99 × 99'} The size of the window should be large enough to contain pixels of the structure being detected and background pixels. A window of the chosen filter size then calculates the average pixel intensity in the window for every pixel in the image and adds the chosen C value to it. If the target pixel original grey scale value is equal to or exceeds this value (average window intensity value + C value), then the target pixel value in the resultant binary image is set to one otherwise it is set to 0. |

## Step 2: Tidy Nucleus Image

The adaptive thresholding stage creates a binary image similar to the one shown below. Holes in the nuclei can be closed by dilation (Dilate) and nuclei can be returned to their original size by the erosion operation (Erode). Any object that touches the boundary of the image (Remove Edge Objects) and any small background objects (Remove Small Objects) can be removed from the image by clicking on them. Click on the Remove Individual Cells check button to allow exclusion of selected objects by clicking on the corresponding outline. Finally, the remaining cells can be counted and labelled by pressing Count Cells.

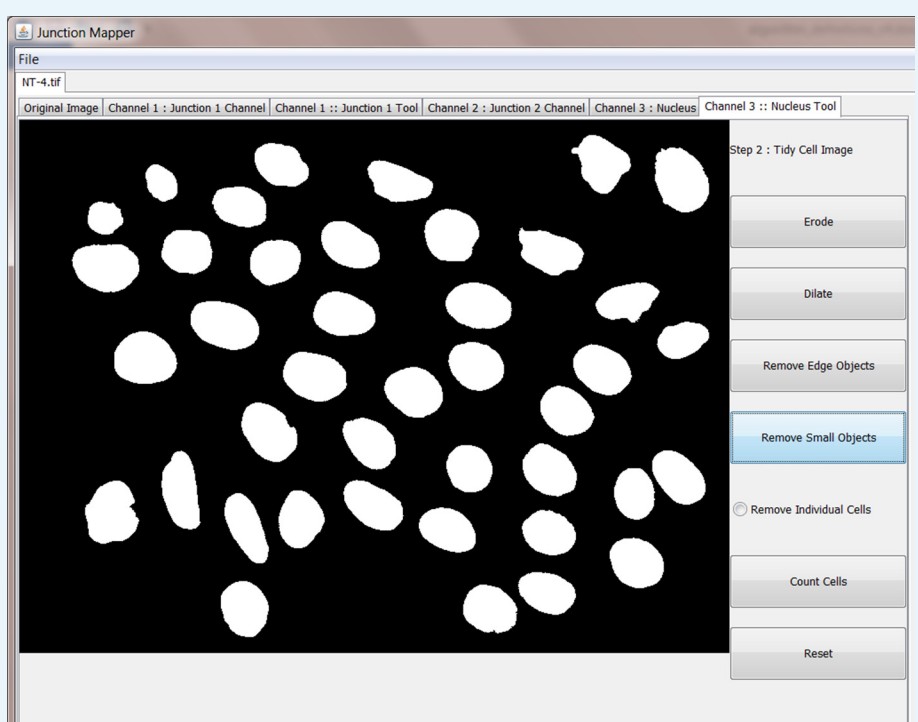

**Appendix 1—figure 12.** Screenshot of Nucleus Tool and the adaptive thresholding parameter setting.

| Control name | Algorithm used |
|---|---|
| Erode | Standard single cycle binary image erosion. |
| Dilate | Standard single cycle binary image dilation. When used in conjunction with erosion can be used to remove holes inside binary objects. |
| Remove Edge Objects | Removes any object that touches the boundary of the image. |
| Remove Small Objects | Removes pixel connected objects from the binary image that are smaller than a user selected threshold. User can choose from the set: {'10','20','30','40','50','100','150','200','250','300','350','400','450','500'} Objects smaller than the selected threshold will be removed from the image |
| Remove Individual Cells | Removes objects from the image that the user selects by clicking on them. |
| Count Cells | Counts the nuclei in the image, assuming each cell has one nucleus. |

## Step 3: Measure Distance between Neighbouring Nuclei

The inter nuclei distances can be measured and output to a spreadsheet in this step – i.e. the Euclidian distance from the centre of a nucleus to the centre of a neighbouring nucleus. The number of neighbouring cells that are measured can be set using the control 'Set # Cells'. The cells to be measured are selected by the user by clicking on them and they appear in the list on the interface below. Cells can be removed from the list by clicking on them again. Once the cells to be measured have been selected, the distances can be saved in an Excel file (Save as Spreadsheet).

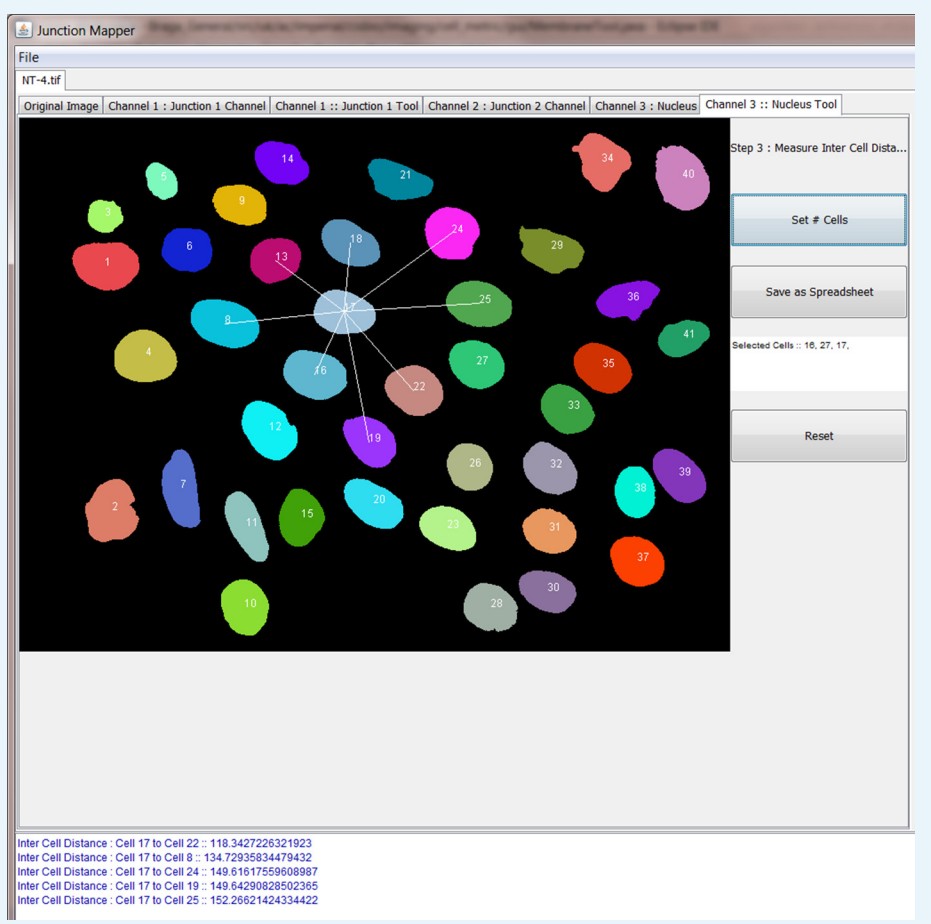

**Appendix 1—figure 13.** Screenshot showing the removal of small objects to identify nuclei in image.

| Control name | Algorithm used |
|---|---|
| Set # Cells | Sets the number of nuclei from neighbouring cells that the distance will be measured for. Item needs to be selected in the pulldown menu. |
| Select Cell | User can select and deselect cells to be measured by clicking on them with the mouse. A list of currently selected cells is displayed on the interface. |
| Save as Spreadsheet | Selected nucleus measurements are calculated and output to a spreadsheet. The central point in a nucleus is calculated as; $(max(x)-min(x))+min(x)$, $(max(y)-min(y))+min(y)$ Where x and y are the coordinates of pixels in a nucleus. We assume that nucleus shapes are roughly symmetrical. Distances calculated are the Euclidian distance from the respective nucleus centres. |

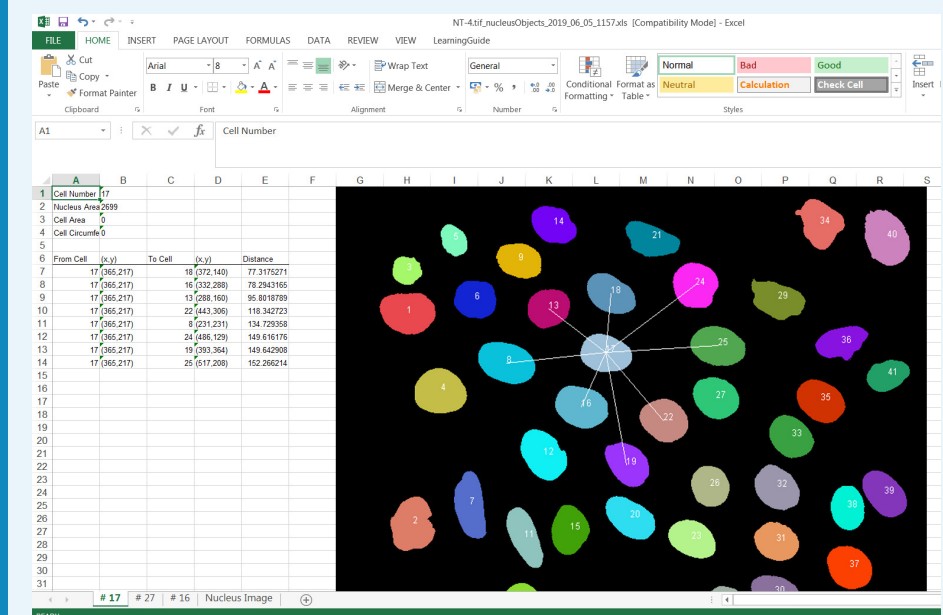

**Appendix 1—figure 14.** Screenshot showing how internuclear distances are measured by the Nucleus Tool.

## Appendix 2

<div style="background:#eaf4fb">

### A - Junction Mapper

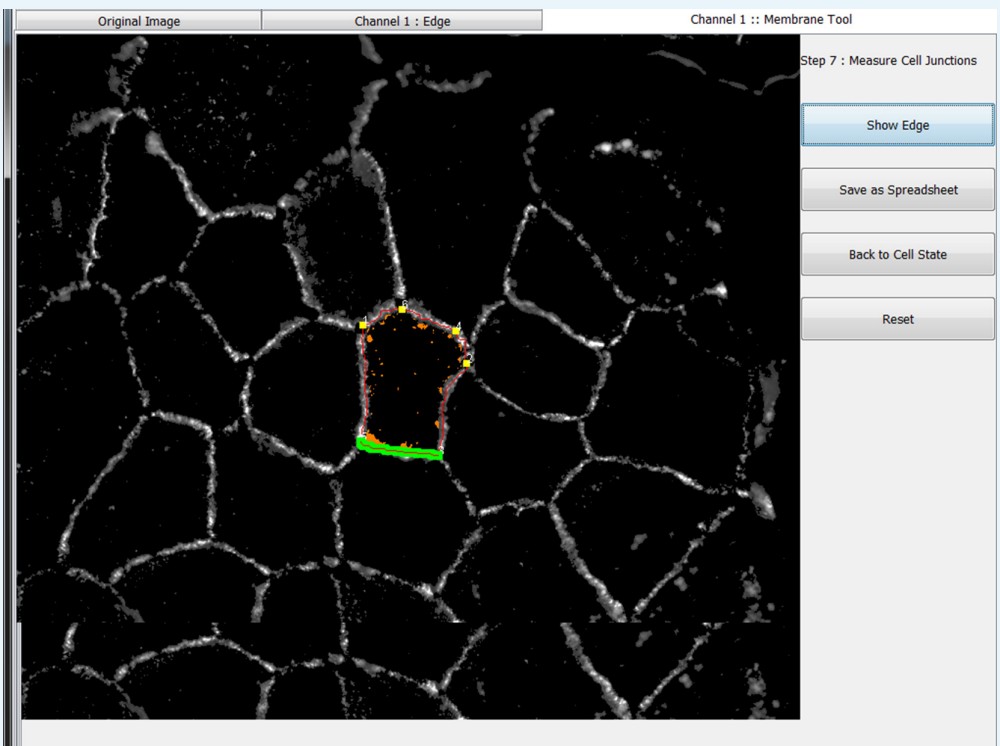

**Appendix 2—figure 1.** Screen shot of the Junction Mapper programme showing the edge channel, define cell edge (skeleton), corner assigned and the dilated area of one junction.

### B - Primary parameters

| # | Name | Units | Description | Details |
|---|------|-------|-------------|---------|
| 1 | Interface Contour | [pixels] | Distance between two corners of the defined cell edge | Algorithm walks along the defined edge from corner to corner and sums the pixel to pixel distance. This is measured by the Euclidian distance between neighbouring pixels along the edge (as diagonally connected pixels distance = approx. 1.4, whilst straight connected pixels = 1). The red line on the image represents the defined cell edge. |
| 2 | Straight-line Interface Length | [pixels] | Straight line distance between two corner points | Euclidian distance between the two corner pixels defined for the edge. The corner pixels are the last pixels at either end of the red line on the image. |

</div>

| 3 | Fragmented Junction Contour | [pixels] | Sum of stained fragments along the single pixel edge | Sum of pixel to pixel distance measured by Euclidian distance between neighbouring pixels along the defined cell edge where the staining intensity exceeds the threshold set. The distance in between individual fragments is NOT calculated. Isolated single above threshold pixels are not included in this measurement. |
| 4 | Dilation Cycles | [unitless] | Number of cycles used to dilate the defined edge | Number of times the binary image dilate algorithm is used to expand the defined edge. Essentially one dilate cycle changes a single pixel line to a three pixels-wide line. Two dilation cycles make the line five pixels-wide, etc. |
| 5 | Interface Area | [pixels$^2$] | Area in pixels of the dilated edge area between two corners | Number of pixels within the dilated edge area defined by two consecutive corners. The dilated area is represented by the green area on the image. Pixels on the red line are also included in this area. |
| 6 | Junction marker 1 Area | [pixels$^2$] | Area covered by cadherin staining (junctional protein) within the interface area | Total number of pixels within the interface area defined in 5 where the junction marker 1 staining has intensities exceeding the threshold set. |
| 7 | Junction marker 1 Intensity | [A.U.] | Sum of cadherin (junctional protein) Intensity within the interface area. | Sum of junction marker 1 intensities in the interface area defined in 6. This measurement only applies to pixels in which marker intensity is above the selected threshold. Pixels below the threshold are set to zero. |
| 8 | Junction Contour | [pixels] | Sum of pixel to pixel distances between the first and last cadherin (junctional protein) pixels along the interface contour | Sum of pixel to pixel distance measured by Euclidian distance between junction marker 1 pixels along the interface contour defined in 1. Distance is measured between the first and last above-threshold pixel, but in between these points all pixels (above and below threshold) are considered; i.e. the distance in-between junction marker 1 fragments (gaps) is also included. |
| 9 | Straight-line Junction Length | [pixels] | Euclidian distance from first to the last pixel of junction marker 1 on the interface contour | Euclidian distance between the first pixel of Junction marker 1 above threshold encountered along the interface contour defined in 1 and the last junction marker 1 pixel above threshold on the edge defined in 1. |

## C - Secondary parameters

| # | Name | Units | Description | Details |
|---|------|-------|-------------|---------|
| 10 | Interface Linearity Index | Ratio [unitless] | Ratio of 'Interface Contour' to 'Straight-line Interface Length' | Measurement 1 / Measurement 2 |
| 11 | Coverage Index | [%] | Ratio of 'Fragmented Junction Contour' to 'Interface Contour' | Measurement 3 / Measurement 1 |
| 12 | Interface Occupancy | [%] | Ratio of Junction marker 1 Area to Interface Area | Measurement 6 / Measurement 5 |
| 13 | Junction marker 1 Intensity per Interface Area | [A.U./pixel$^2$] | Ratio of 'Junction marker 1 Intensity' to 'Interface Area' | Measurement 7 / Measurement 5 |
| 14 | Cluster Density | [A.U./pixel$^2$] | Ratio of 'Junction marker 1 Intensity' to 'Junction marker 1 Area'. | Measurement 7 / Measurement 6 |
| 15 | Junction Linearity Index | Ratio [unitless] | Ratio of 'Junction contour' and 'Straight-line junction length' | Measurement 8 / Measurement 9 |

