## [Decision Letter]

**Acceptance summary:**

This manuscript provides a novel tool to quantify different parameters of mammalian intercellular junctions, and as such, is aptly called junction mapper. As pointed out by the authors in the Introduction of the manuscript quantification of different mammalian junctional parameters is often rather time consuming as for many programs finding borders and, often, correcting borders, can until now mostly be done manually. This tool will be useful to researchers in many fields for quantifying cell junctions.

**Decision letter after peer review:**

[Editors’ note: the authors were asked to provide a plan for revisions before the editors issued a final decision, this was subsequently approved by the editors. What follows is the editors’ letter requesting such plan.]

Thank you for sending your article entitled "Junction Mapper: quantitative analysis to decipher cell-cell contact phenotypes" for peer review at *eLife*. Your article is being evaluated by three peer reviewers, and the evaluation is being overseen by a Reviewing Editor and Anna Akhmanova as the Senior Editor.

Given the list of essential revisions, including new experiments, the editors and reviewers invite you to respond within the next two weeks with an action plan and timetable for the completion of the additional work. We plan to share your responses with the reviewers and then issue a binding recommendation.

While the reviewers recognized the benefits of the tool to analyze junctions, the reviewers felt that the manuscript lacked details regarding the algorithm testing and methods. Furthermore, the reviewers also would like to see whether this tool can be used on mammalian junctions to rigorously test whether this will be broadly used by the field of epithelial junction biologists.

Reviewer #1:

This very nice manuscript provides a novel tool to quantify different parameters of mammalian intercellular junctions, and as such, is aptly called junction mapper. As pointed out by the authors in the Introduction of the manuscript quantification of different mammalian junctional parameters is often rather time consuming as for many programs finding borders and, often, correcting borders, can until now mostly be done manually. Automated analysis is not trivial and in general requires a strong background in image bioinformatics/analysis or access to a core facility that provides such analysis. The exciting point of the work presented here is that this program provides a general and very accessible platform to quantify junctions in a semi-automatic manner to obtain multiple parameters. Moreover, proper detection of fragmented junctions is an even more complex problem which the authors have apparently solved. The program provides reasonable output parameters and the detection quality and sophistication of junctional staining appears to be outstanding judged by the data presented on both epithelial junctions as well as the already much harder endothelial junctions. Finally, the experiments that were performed convincingly demonstrate the applicability of the program and its sensitivity of detecting minor and specific alterations. Taken together, this manuscript provides convincing and exciting evidence that junction mapper is a great open software to quantitatively analyze junctions, and that allows laboratories not specialized in junctional imaging to also more precisely analyze and assess junctional phenotypes.

I have two comments that need to be addressed:

1) The last author is a well-known specialist on keratinocytes, which in vivo and in culture form multi-layered epithelia upon prolonged induction of differentiation. Although the authors have used keratinocytes, it seems they only used early time points after differentiation when these cells did not form multilayers yet. The multi-layering provides an extra complexity to the proper quantification of junctions. Can the program deal with this complexity? If so, would be very nice to include an example. If not, please also more stringently then discuss the limitations of the program.

2) One important point does need to be clarified. It is not exactly clear from the figures, legends or Materials and methods how the statistics were done. The authors name all the appropriate tests but it is not clear e.g. whether means of technical replicates or the collective number of junctions were used for statistical test. Clearly, the high number of junctions that can be analyzed here is an advantage. However, the variance between experiments is not shown and it is not clear whether the statistics take that into account.

Reviewer #2:

1) This manuscript describes a novel semiautomated software, 'Junction Mapper,' for analyzing cell junctions on parameters such as defining the junctional interphase, intensity, and length. Currently, analyzing cell junctions is often done manually which can be time consuming and laborious, even when using automated software, it is usually tailored to a specific cell type or unable to automatically distinguish nuances of perturbed or fragmented junctions. The authors evaluate their software's performance in several cell types, and in multiple conditions to evaluate its ability to detect a range of phenotypes (from severe to mild). This manuscript presents a new tool that address these significant problems and has the potential to be widely applied. However there are numerous flaws and oversights in this presentation that significantly dampen my enthusiasm. There is also a lack of rigor in evaluating the algorithms as presented which makes it difficult to assess the utility of the software thereby raising a major concern.

2) First, the availability of the software and test data sets is not clear. Nor is the computing language or platform that the software is written in. This is a major oversight. Second, the microscopy needs to be fully described. There are not scale bars on images. Are these max intensity projections, single confocal slices, widefield images, deconvovled? What is the resolution? The quality of the input will impact the ability of the algorithms to function and this needs to be defined.

3) How do different user selected parameters impact the output from the program? Since the skeleton can be manually adjusted, does the specific skeleton have an impact on the results? These factors need to be considered and quantified for their impact on the final measurements.

4) Input parameters need to be discussed – dilation and threshold level are important parameters to disclose for each analyzed data set. How do these impact final measurements?

5) The process of interactive editing, blurring, and sharpening is not shown nor well described. Is this process done manually on every image? Can it be automated for a data set? Seeing these steps, including when manual adjustment is required would be helpful. The reader has no feel for how often user intervention is required (every cell, every image, every border?).

Reviewer #3:

The manuscript describes the development of software for quantifying the phenotype of cell-cell contacts from high resolution images. The authors have been able to produce an impressive data set with the tool. However, the new tool is not validated sufficiently, and the methods are not adequately described. This is particularly important, as by the studying design, the biological ramifications of the analysis were not probed.

1) In the Introduction (third paragraph), the authors make the claim that the software developed for analysis of Drosophilia or nematode epithelia are not applicable to mammalian cells. Image analysis algorithm performance are typically based of the nature of the structures to be analyzed and the quality of data. There is no reason for these parameters to be fundamentally different in all mammalian cells. If the authors, are trying to say that tools for analysis of epithelial cells may not be directly applicable to other cells types or in all possible treatments (points made later), they should be more explicit here. Also, the (Held et al., 2011) does not describe the analysis of cell-cell contacts and does not seem to support the authors point.

2) The development of a new algorithm for image analysis is typically associated with several key endeavors. First is validation. Validation must be performed on data where the ground truth is known. For a "first-of-kind" algorithm this is often associated done with simulated data or data that was analyzed by hand. By this standard definition, the Junction Mapper is not validated. This must be corrected for the manuscript to be considered for publication. It seems that the data shown in Figure 6 may be appropriate for this task, as the data was previously analyzed by hand? This reviewer notes that citations within the manuscript are validated in this manner (i.e. Held et al., 2011). Also, the section in the Materials and methods described "Software validation" actually discusses exclusion criteria and naming procedures.

3) Additionally, there is typically an analysis of when the algorithm begins to fail. This is often done in terms of data with varying signal to noise. The limitations of this of the Junction Mapper are not tested. While it would be preferred that a formal analysis of the performance of the algorithm be performed, at a minimum image quality (which is often reported as signal to noise ratios) requirements (or at least typical image characteristics) should be provided so that users know the criteria for the suitability of the algorithm are met being attempting to use it.

4) The Materials and methods also lack sufficient detail. According to the manuscript, the Junction Mapper is a novel stand alone software. This means specific algorithms for background estimation, skeletonization, segmentation, and parameter calculation were used. Precise definitions (including formulae) or citations to the utilized algorithms need to be provided. Also, formulae describing how the parameters were calculated need to be added. Also, how corners were identified needs to be explained in greater detail. If corners are manually detected, then it is improper to state the Junction Mapper detected them. Additionally, are all junctions analyzed at once or individually? If different segmentation parameters are used on different junctions, are the results actually directly comparable? This reviewer also notes that the required details are not available in the often-cited previous work of Erasmus et al., 2016. This is understandable as that was a highly biological study, but that reference should not be used to described methodological details.

5) In several places the authors make claims that junctions that are straighter are more tensed. This is not true, as the junctions could also be stiffer, and therefore deform less. As no measurements of junctional tension were made, these statements should be removed.

6) The algorithm seems to assume constant width for the junctions. This will likely lead to substantial errors in area estimations. Have the authors given thought to this issue? Are the errors inconsequential? Can the algorithm be readily updated to avoid this artefact?

[Editors' note: further revisions were requested prior to acceptance, as described below.]

Thank you for resubmitting your work entitled "Junction Mapper is a novel computer vision tool to decipher cell-cell contact phenotypes" for further consideration at *eLife*. Your revised article has been favorably evaluated by Anna Akhmanova as the Senior Editor, a Reviewing Editor, and three reviewers.

The manuscript has been improved but there are some remaining issues that need to be addressed before acceptance, as outlined below:

Reviewer #1:

The authors took a lot of effort to address the points that were raised by the reviewers. I do not have the proper expertise to judge whether the authors have sufficiently addressed the points of reviewers 2 and 3.

1) In response to point 2, the authors now describe which statistical tests are used and clarify for which figure panels they used biological or technical replicates. The authors now state that the statistics and the tests used are in Supplementary file 5. Unfortunately, I was unable to find Supplementary file 5 in the present submission, making it impossible whether the statistics were done properly. Importantly, the authors still do not clearly state in the legends and/or Materials and methods the number of replicates on which the data shown are based on. Throughout the manuscript, each measured junction is called a sample and the number of analyzed junctions are now provided in the figures. However, for the statistics the authors should provide information on the number of replicates, meaning number of independent experiments or biological samples, in the figure or legend to clearly show that differences and significance levels are due to the high number of junctions but not because of experimental variation.

2) For siRNA experiments the section on statistical analysis says that only one control and knockdown group was analyzed. If this is not a misunderstanding, then even high number of analyzed junctions from one experiment do not allow strong conclusions as experimental variation upon knockdown of CIP4, EEFA1A or VAV2 has not been assessed. For physiological relevant conclusions one would expect a set of at least 3 independent experiments. I do understand that the authors want to test their very interesting tool for which they provide many different experimental tests. However, the authors should then acknowledge and tune down mechanistic conclusions. For experiments concerning HUVEC cells, there is no information on the number of biological replicates on which the data are based.

3) Thus, at present the way the data are presented now, provides nice evidence that the program allows for quantification of high number of junctions to potentially detect small differences. If the authors want to use the experiments shown in the manuscript to validate the usefulness as well as limits of the software, this approach with limited number of repetitions seems valid but then this has to be made more clearly throughout the manuscript. Not all of the data sets are sufficiently comprehensive to univocally demonstrate a biological relevant junctional outcome upon treatment or knockdown and this should be acknowledged. Having said this, I feel if the authors were able to address the concerns of reviewer 2 and 3, they should be provided the opportunity to properly address and include in the manuscript the comments above.

Reviewer #2:

Overall this tool has the potential to be useful for researchers, and the authors have now included many more rigorous definition of the method and treatments of the parameters both in definition and in comparison of data processed in different ways. This strengthens the manuscript. However, I am still not sure this is sufficient to fully describe the bounds of the junction mapper approach.

1) Most scientific conclusions as presented come from comparisons of values in control/disrupted. When looking at effects of user inputs the graphs are shown for each treatments individually but what are the changes with different thresholds/dilations between treatments (i.e. Figure 3—figure supplements 2 and 3)? Would different biological conclusions be drawn if you push the dilation or threshold "too far"?

2) Authors point out a central issue and why this analysis is challenging – If there is no staining at a cell corner then where the program or user places the junction is arbitrary. Without a second marker of plasma membrane rather than junction this will always be an issue these measurements regardless of what automated algorithm is deployed or with user definition.

3) The authors show secondary characteristics are more robust to error in corner selection and that individual users get same direction of changes but not same absolute values, which does show the value added here. This should be emphasized in the Discussion.

4) For the noise quantification, there should be a metric for measurement accuracy, or deviation from the ground truth. Guidance for user on minimum S/N needed for robust quantification. There is also no information on type of noise added (shot noise, dependent on pixel intensity is dominant noise in most microscopy images and is recommended here). The goal would be for a user to know if a junction has sufficient S/N to be analyzed.

5) This leads me to the quantification of the type "ee" boarders in the H-Ras^G12V^ overexpressing cells. There appears to be greatly reduced e-cad staining at the boarders (it is difficult to see by eye any in the examples shown). In this case, how are the interfaces defined? Does this lower signal to noise negatively impact the program? How confident should we feel about comparing images with different S/N levels?

Reviewer #3:

The manuscript is greatly improved and nearly ready for publication.

---

## [Author Response]

[Editors’ notes: the authors’ response after being formally invited to submit a revised submission follows.]

Reviewer #1:[…] I have two comments that need to be addressed:1) The last author is a well-known specialist on keratinocytes, which in vivo and in culture form multi-layered epithelia upon prolonged induction of differentiation. Although the authors have used keratinocytes, it seems they only used early time points after differentiation when these cells did not form multilayers yet. The multi-layering provides an extra complexity to the proper quantification of junctions. Can the program deal with this complexity? If so, would be very nice to include an example. If not, please also more stringently then discuss the limitations of the program.

There could be adaptations to quantify en face confocal Z-sections of a stratified epithelia. However, the complexity of stratified epithelium images (en face or transversal sections (see Figure 3A from Sevilla et al., 2007) means that a potential analysis using Junction Mapper will not be meaningful. For example, it will be very much dependent on the angle of the section, variations in cell sizes and the amount of incomplete outlines of individual cells, which makes comparison across samples difficult. It also remains to be tested whether Junction Mapper can quantify junctions in pseudo-stratified epithelia. This information and limitations were added to the Discussion.

2) One important point does need to be clarified. It is not exactly clear from the figures, legends or Materials and methods how the statistics were done. The authors name all the appropriate tests but it is not clear e.g. whether means of technical replicates or the collective number of junctions were used for statistical test. Clearly, the high number of junctions that can be analyzed here is an advantage. However, the variance between experiments is not shown and it is not clear whether the statistics take that into account.

Apologies for the oversight and unclear text. The results are mostly from technical replicates: 5-10 images were analysed per sample and all junctions from cells in each image were quantified (except those excluded by the quality control criteria). Each point on graphs represents a measurement from one junction. The numbers ('n =') cited include all junctions analyzed (pooled from all available images of a given sample). For some experiments (Src expression and endothelial cells), biological replicates were analysed separately. If the pattern of the parameters were similar across biological replicates, the data was pooled all together and shown in the graph. A new supplementary table contains the statistics performed and where appropriate, the variance (Supplementary file 5). Clarifications on which data are from a technical or biological replicate were added in Supplementary file 1 and in the figure legends.

Reviewer #2:[…]1) This manuscript presents a new tool that address these significant problems and has the potential to be widely applied. However there are numerous flaws and oversights in this presentation that significantly dampen my enthusiasm. There is also a lack of rigor in evaluating the algorithms as presented which makes it difficult to assess the utility of the software thereby raising a major concern.

We thank the reviewer for the valid points made. The parameter explanations and methodology could have been made clearer and further validation presented. We have now provided extensive validation of the program in several supplementary figures, main figures and tables (please see those outlined below).

2) First, the availability of the software and test data sets is not clear. Nor is the computing language or platform that the software is written in. This is a major oversight. Second, the microscopy needs to be fully described. There are not scale bars on images. Are these max intensity projections, single confocal slices, widefield images, deconvovled? What is the resolution? The quality of the input will impact the ability of the algorithms to function and this needs to be defined.

The reviewer has a point and we are sorry for the oversight. The test data sets are from our own labs, unpublished data and from published papers; i.e. from Erasmus et al., 2016 – these are referred to in the text. The software was developed using Java and is available as an open access standalone, downloadable format (information added in the subsection “Software development”). A new supplementary table containing detailed information of the acquisition of images for each dataset was added (Supplementary file 1). Explanations of what critically affects the analyses and the limitations of the software were written in the Materials and methods and Results(please see also point 3, 4 below and reviewer 3 points 2, 3, 4 and 6).

3) How do different user selected parameters impact the output from the program? Since the skeleton can be manually adjusted, does the specific skeleton have an impact on the results? These factors need to be considered and quantified for their impact on the final measurements.

The user can control four distinct steps. The first two (skeleton and corner identification) are done automatically but can be revised by the user. While it is important that the skeleton overlays the junctional marker staining, there is some tolerance on the precision of skeleton outline, because there is a dilation step to define the ROI.

The other two user-controlled settings are to choose the dilation value (to encompass the whole width of the staining along the skeleton) and the thresholding levels (to keep noise in the cytoplasm at a minimum, without information loss at junctions). These settings are kept constant for all images of a given experiment. Explanations are added to the text to clarify the relevance of the user-defined settings (Results and in “Figure 2—figure supplement 2”).

4) Input parameters need to be discussed – dilation and threshold level are important parameters to disclose for each analyzed data set. How do these impact final measurements?

Thanks for the suggestion. In the revised manuscript, two new supplementary figures (Figure 3—figure supplement 1 to 2) demonstrate the effects of varying dilation and thresholding on the primary parameters (length and intensity). A supplementary table was added with the information of how each sample was analysed (dilation and thresholding settings; Supplementary file 1).

5) The process of interactive editing, blurring, and sharpening is not shown nor well described. Is this process done manually on every image? Can it be automated for a data set? Seeing these steps, including when manual adjustment is required would be helpful. The reader has no feel for how often user intervention is required (every cell, every image, every border?).

We agree with the reviewer. The process is done in each image because of image-to-image variability and the extent of junction disruption when stimulated. We addressed the concerns in two ways: (i) a new supplementary figure shows the precise steps to obtain the skeleton and corners (Figure 2—figure supplement 2) and (ii) instructions on how the software operates are included in a custom-made video downloadable package of Junction Mapper from a dedicated website.

Reviewer #3:[…]1) In the Introduction (third paragraph), the authors make the claim that the software developed for analysis of Drosophilia or nematode epithelia are not applicable to mammalian cells. Image analysis algorithm performance are typically based of the nature of the structures to be analyzed and the quality of data. There is no reason for these parameters to be fundamentally different in all mammalian cells. If the authors, are trying to say that tools for analysis of epithelial cells may not be directly applicable to other cells types or in all possible treatments (points made later), they should be more explicit here.

This is a misunderstanding. What the referred paragraph is meant to say is that algorithms that work fine to identify cell borders (i.e. skeleton) in *Drosophila* or *C. elegans* epithelia struggle to perform well in mammalian epithelia. This is because mammalian junctions are thinner, more irregular and have higher signal to noise ratio than those of in vivo images of invertebrate epithelia. The Junction Mapper parameters (novel and established) are universal and are applicable to epithelia from different organisms, cell types and following different treatments.

Also, the (Held, et al., 2011) does not describe the analysis of cell-cell contacts and does not seem to support the authors point.

Apologies, this reference is incorrect as it refers to different ways to identify cells in a monolayer.

2) The development of a new algorithm for image analysis is typically associated with several key endeavors. First is validation. Validation must be performed on data where the ground truth is known. For a "first-of-kind" algorithm this is often associated done with simulated data or data that was analyzed by hand. By this standard definition, the Junction Mapper is not validated. This must be corrected for the manuscript to be considered for publication. It seems that the data shown in Figure 6 may be appropriate for this task, as the data was previously analyzed by hand? This reviewer notes that citations within the manuscript are validated in this manner (i.e. Held et al., 2011). Also, the section in the Materials and methods described "Software validation" actually discusses exclusion criteria and naming procedures.

The reviewer is correct. Many of the parameters are novel and have not been used previously. Unfortunately, data shown in Figure 6 has been quantified by another thresholding method, using the intensity of the whole image as output (i.e. all cells in the image).^[2]^ The raw data obtained is thus not comparable, as the results are from the intensity of all junctions in the whole image corrected for the total image area.^[2]^ The raw data from Junction Mapper output measure values per junction corrected for its respective interface area. Nevertheless, in an experiment, when the percentage of rescue is computed with either our previous and current method, similar results are obtained: a mild reduction of E-cadherin intensity at junctions compared to controls after depletion of TRIP10 or VAV2 (20-30%) while EEF1A shows a 50% increase in E-cadherin intensity at junctions (Figure 6).^[2]^ We provided validation in several ways and revised the text accordingly:

- Length-based and area-based parameters are measured as predicted, i.e. with increasing shorter values for the interface, junctions and fragmented junction staining (new Figure 3B).

- Dilation manipulation interferes with area-based measurements, but not length-based parameters Figure 3—figure supplement 2).

- Increasing levels of thresholding interferes with intensity and pixel area as well as the length of fragments of junction markers (Figure 3—figure supplement 3).

- The validation of the secondary parameter Coverage Index by manual quantification as described in our previous paper (i.e. Supplementary file 2).^[3]^

- Comparison of quantification by an independent user (new Supplementary file 3 and Figure 7—figure supplement 1; please see points 3, 4 and 6 below).

3) Additionally, there is typically an analysis of when the algorithm begins to fail. This is often done in terms of data with varying signal to noise. The limitations of this of the Junction Mapper are not tested. While it would be preferred that a formal analysis of the performance of the algorithm be performed, at a minimum image quality (which is often reported as signal to noise ratios) requirements (or at least typical image characteristics) should be provided so that users know the criteria for the suitability of the algorithm are met being attempting to use it.

We thank the reviewer for the suggestion. Additional explanations were added in the Results and a new supplementary figure provide validation of signal-to-noise ratios (new Figure 2—figure supplement 3).

4) The Materials and methods also lack sufficient detail. According to the manuscript, the Junction Mapper is a novel stand alone software. This means specific algorithms for background estimation, skeletonization, segmentation, and parameter calculation were used. Precise definitions (including formulae) or citations to the utilized algorithms need to be provided. Also, formulae describing how the parameters were calculated need to be added. Also, how corners were identified needs to be explained in greater detail. If corners are manually detected, then it is improper to state the Junction Mapper detected them.

The software uses algorithms that are mostly available open access, with some added innovation. The novelty of the Junction Mapper is in the integration of distinct measurements, calculations of new parameters and consolidation of all parameters in a single system. The mathematical definitions of the parameters, the innovation of the automatic corner setting, its correction by users and the calculation of fragmented regions of contacts are now described in a more comprehensive explanation of the software (new Figure 2—figure supplement 2). This information was added to the Materials and methods section.

Additionally, are all junctions analyzed at once or individually? If different segmentation parameters are used on different junctions, are the results actually directly comparable?

Once the skeleton, corners and dilation are set, all junctions in the ROI are quantified simultaneously at a click, and all measurements and parameters exported to an excel file automatically. For results to be comparable, images from an experiment with various treatments would need to be analysed with the same conditions (dilation, thresholding, etc). This information was added to the revised text (Results) and in new Figure 2—figure supplement 2.

This reviewer also notes that the required details are not available in the often-cited previous work of Erasmus et al., 2016. This is understandable as that was a highly biological study, but that reference should not be used to described methodological details.

The reviewer has a point and it needs to be better clarified in the text. The work on Erasmus et al. defines the obtention of the E-cadherin mask as a tool to calculate the intensity specifically around junctions as% thresholded area of the whole image. It also defines the Ecadherin mask to subtract an ROI on an image of a distinct marker (i.e. F-actin), so that only the signal localized at contacts is considered. Junction Mapper builds from this work and improves with novel quantification tools (corners, length and area), calculation of fragments at junctions and a variety of novel primary and secondary parameters. These points were clarified in the text (Results and Materials and methods).

5) In several places the authors make claims that junctions that are straighter are more tensed. This is not true, as the junctions could also be stiffer, and therefore deform less. As no measurements of junctional tension were made, these statements should be removed.

Thanks for point out this over-interpretation – we have indeed shown no data to support either case (tension or stiffness). The reviewer is correct that junctions could also be stiffer. However, the statement is based on numerous papers that: (i) using laser ablation show a striking recoil of junctions, which is abolished by different treatments to inhibit Rho pathway ^[4]^, (ii) show weaker, undulated contacts upon enhancing signalling from Cdc42 GTPase ^[1]^, (iii) measurement of tension at junctions with biosensors^[5]^ and (iv) the wiggly morphology of junctions observed in tumour cells versus in vivo in epithelial tissues. The text was revised throughout as requested.

6) The algorithm seems to assume constant width for the junctions. This will likely lead to substantial errors in area estimations. Have the authors given thought to this issue? Are the errors inconsequential? Can the algorithm be readily updated to avoid this artefact?

This is a misunderstanding. The variable width of staining at junctions can be fully captured by the dilation step (user defined) and will depend on the amplification of the image. The dilation steps increase the numbers of pixels at each side of the skeleton line (range 0 to 9 pixels) and determines automatically the area in which quantification will be done (ROI). The influence of varying the dilation for different parameters is now shown in new Figure 3—figure supplement 2. The dilation value is manually set up and kept constant for each dataset (i.e. H-Ras, Rac1, etc). We provided the dilation values in new Supplementary file 1 along with the details for final thresholding and image dataset characteristics).

References:

1) Otani, T., et al., Cdc42 GEF Tuba regulates the junctional configuration of simple epithelial cells. J. Cell Biol., 2006. 175(1): p. 135-146.

2) Erasmus, J.C., et al., Defining functional interactions during biogenesis of epithelial junctions. Nature Commun., 2016. 7: p. 13542.

3) Lozano, E., et al., PAK is required for the disruption of E-cadherin adhesion by the small GTPase Rac. J. Cell Sci., 2008. 121(Pt 7): p. 933-938.

4) Priya, R., et al., Feedback regulation through myosin II confers robustness on RhoA signalling at E-cadherin junctions. Nature Cell Biol., 2015. 17: p. 1282-1293.

5) Acharya, B.R., et al., Mammalian Diaphanous 1 Mediates a Pathway for E-cadherin to Stabilize Epithelial Barriers through Junctional Contractility. Cell Rep., 2017. 18(12): p. 2854-2867.

[Editors' note: further revisions were requested prior to acceptance, as described below.]

Reviewer #1:[…]1) In response to point 2, the authors now describe which statistical tests are used and clarify for which figure panels they used biological or technical replicates. The authors now state that the statistics and the tests used are in Supplementary file 5. Unfortunately, I was unable to find Supplementary file 5 in the present submission, making it impossible whether the statistics were done properly. Importantly, the authors still do not clearly state in the legends and/or Materials and methods the number of replicates on which the data shown are based on. Throughout the manuscript, each measured junction is called a sample and the number of analyzed junctions are now provided in the figures. However, for the statistics the authors should provide information on the number of replicates, meaning number of independent experiments or biological samples, in the figure or legend to clearly show that differences and significance levels are due to the high number of junctions but not because of experimental variation.

The information was present in the legends but we acknowledge that it was not very clear. We now revised each relevant figure legend and state whether the data are technical or biological replicates. We changed the legends to state “number of junctions” instead of “number of samples”. The text was also modified to clarify the significance due to the high number of junctions (subsection “Distinct oncogenes trigger different patterns of junction disruption in epithelia”) and revised the correct citations to supplementary materials to be fully compliant with *eLife* requirements.

2) For siRNA experiments the section on statistical analysis says that only one control and knockdown group was analyzed. If this is not a misunderstanding, then even high number of analyzed junctions from one experiment do not allow strong conclusions as experimental variation upon knockdown of CIP4, EEFA1A or VAV2 has not been assessed. For physiological relevant conclusions one would expect a set of at least 3 independent experiments. I do understand that the authors want to test their very interesting tool for which they provide many different experimental tests. However, the authors should then acknowledge and tune down mechanistic conclusions. For experiments concerning HUVEC cells, there is no information on the number of biological replicates on which the data are based.

We thank the reviewer for the suggestion and we endeavour to eliminate any over-interpretation. We are acutely aware that the above distinction should be made very clear to the readers, and are happy to clarify further. While there were caution notes in the text, we have revised the text to explain this point better, tone down the conclusions and extensively altered the Discussion section to focus more on Junction Mapper properties, advantages and caveats.

3) Thus, at present the way the data are presented now, provides nice evidence that the program allows for quantification of high number of junctions to potentially detect small differences. If the authors want to use the experiments shown in the manuscript to validate the usefulness as well as limits of the software, this approach with limited number of repetitions seems valid but then this has to be made more clearly throughout the manuscript. Not all of the data sets are sufficiently comprehensive to univocally demonstrate a biological relevant junctional outcome upon treatment or knockdown and this should be acknowledged. Having said this, I feel if the authors were able to address the concerns of reviewer 2 and 3, they should be provided the opportunity to properly address and include in the manuscript the comments above.

Please see our reply to point 2 above and to the other reviewers below.

Reviewer #2:[…]1) Most scientific conclusions as presented come from comparisons of values in control/disrupted. When looking at effects of user inputs the graphs are shown for each treatments individually but what are the changes with different thresholds/dilations between treatments (i.e. Figure 3—figure supplements 2 and 3)? Would different biological conclusions be drawn if you push the dilation or threshold "too far"?

We understand the reviewer point. In the analysis, we keep the threshold at a minimum to avoid loss of signal at junctions (subsection “Software development”, fourth paragraph). Very high thresholding levels are clearly not appropriate for the analyses, as too much signal is lost (i.e. threshold 100-150, Figure 3—figure supplement 3). Similarly, very high dilation settings may not detect any further increase in measurement values in stimulated samples, but not in controls, in some (Figure 3—figure supplement 2A-D), but not all experiments (Figure 3—figure supplement 2E-H).

As shown in our validation experiments, different users, using distinct dilation and thresholding settings obtain different absolute values for the parameters using Junction Mapper (Figure 7—figure supplement 1; Supplementary file 3). However, the conclusions are statistically similar when compared with controls quantified under the same conditions. As the settings are applied to both control and treated images, we expect that the results are not changed overall. Higher threshold levels will remove intensity at junctions and cytoplasm in both control and treated groups. Conversely, we expect that by increasing dilation much wider than the junctional staining, the background or noise will be computed as junctional signals for all samples (control and treated).

2) Authors point out a central issue and why this analysis is challenging – If there is no staining at a cell corner then where the program or user places the junction is arbitrary. Without a second marker of plasma membrane rather than junction this will always be an issue these measurements regardless of what automated algorithm is deployed or with user definition.

The reviewer is correct: a membrane or cell marker is very helpful to guide corner positioning. This can be done using cytoskeletal staining (i.e. F-actin) or a marker that labels the whole cell; i.e. Cell Mask. The above point is stressed in the third paragraph of the subsection “Software development”.

3) The authors show secondary characteristics are more robust to error in corner selection and that individual users get same direction of changes but not same absolute values, which does show the value added here. This should be emphasized in the Discussion.

Thanks for the suggestion – this information is now discussed in the subsection “User-bias validation” and in the first paragraph of the Discussion.

4) For the noise quantification, there should be a metric for measurement accuracy, or deviation from the ground truth. Guidance for user on minimum S/N needed for robust quantification. There is also no information on type of noise added (shot noise, dependent on pixel intensity is dominant noise in most microscopy images and is recommended here). The goal would be for a user to know if a junction has sufficient S/N to be analyzed.

We thank the reviewer for the recommendation. We performed the determination of peak signal to noise ratio (PSNR) in the original images compared to those where noise was artificially added using FIJI. Our quantification showed that a PSNR above 21.72dB +/- 0.67 is optimal and generally measured in the original images (cardiomyocytes and keratinocyte samples). Reducing the PSNR values to below 16.00dB (epithelia) or 18.00dB (cardiomyocytes) severely compromises the performance of the software. This information is now added to the subsection “Software validation” and in the Results. The PSNR values were added to each modified image (original and modified, Figure 2—figure supplement 3).

5) This leads me to the quantification of the type "ee" boarders in the H-Ras^G12V^ overexpressing cells. There appears to be greatly reduced e-cad staining at the boarders (it is difficult to see by eye any in the examples shown). In this case, how are the interfaces defined? Does this lower signal to noise negatively impact the program? How confident should we feel about comparing images with different S/N levels?

As stated in point 2 above, for severely perturbed junctions, the junctional remains, staining of neighbouring cells and with a second marker are helpful to delineate the borders. The visualization is also much better in the computer screen. Of note is that we find that samples that are severely disrupted are not easily rescued, as the perturbation goes too far. The experimental design is best when the expression/stimulus is carefully optimized to significantly disrupt in a quantifiable manner, but not completely abolish the junctional staining. The potential of the Junctional Mapper software is to be able to analyse high numbers of junctions and then obtain the sample variability in a statistically significant manner.